# Breaking Static Paradigms: A Mutual Evolution Framework for Edge-Cloud Model Collaboration

## Abstract

To simultaneously achieve high performance and low latency, the paradigm of *edge-cloud* model collaboration, where Large Language Models (LLMs) are deployed on the powerful cloud and Small Language Models (SLMs) on the resource-limited edge devices, has garnered great attention recently. However, a key limitation of current edge-cloud architecture is its static nature, which hinders the dynamic integration of new knowledge. More specifically, existing methods typically update the system by directly retraining the cloud-based LLM with edge-side newly collected data, which not only increases communication overhead but also neglects available computing power and data accessibility on edge devices. To tackle this challenge, we propose a novel mutual Evolution framework for edge-cloud model Collaboration called **CoEvo** that enables both cloud-side LLM and edge-side SLMs to update with new knowledge continuously. The cloud-based LLM can enhance edge-side SLMs through credible Chain-of-Thought (CoT) based knowledge distillation to improve its general understanding capabilities. Once the edge-side SLMs collect new domain-specific knowledge and optimize themselves locally, they will specifically enhance the cloud-based LLM via a credible probability matrix predicted on a few samples without uploading all raw data. Through this mutual evolution, the system can achieve continual optimization of the cloud and edge-side models and promote real-world deployments. Experimental results demonstrate a considerable performance gain of our edge-side SLMs against existing methods on the target dataset, with the cloud-side LLM also achieving a notable improvement over the base model.

## 1 Introduction

Large language models (LLMs) such as the GPT series (Achiam et al., 2024) and DeepSeek R1 (Guo et al., 2025) have been extensively adopted across diverse domains, substantially improving operational efficiency and fostering innovation in a wide range of industries (Saha et al., 2025). To support the practical deployment of these models, the edge–cloud collaborative architecture (Wang et al., 2024b) has emerged as a critical paradigm. This architecture capitalizes on the abundant computational resources of cloud servers to maintain large-scale LLMs, while simultaneously deploying lightweight small language models (SLMs) on resource-constrained edge devices (Tian et al., 2024). By integrating the powerful inference capabilities of LLMs with the lightweight and efficient deployment of SLMs (Zhao et al., 2024), the edge–cloud architecture achieves an effective balance between performance and efficiency. In particular, techniques such as LLM-guided inference enable SLMs (Liu et al., 2024) to inherit knowledge and enhance their reasoning abilities, all while sustaining low latency and minimizing computational overhead. This collaborative paradigm thus provides a promising pathway for scaling LLM applications to real-world environments.

In recent years, researchers have made notable progress in improving the inference performance of edge-cloud architectures. By leveraging maintained external knowledge bases, language models enhance overall performance on both edge and cloud sides through retrieval-augmented generation (RAG) (Lewis et al., 2020; Liu et al., 2025; Qin et al., 2025). To avoid substantial external storage and query overhead during the inference stage, some studies have utilized external tools (Zhuang et al., 2023; Yuan et al., 2024a) such as search engines and compilers to assist language models in

solving practical problems. (Chen et al., 2024a) utilizes detection algorithms to filter data, retaining routine data processing on the edge side while uploading only critical data to the cloud, thereby maintaining the inference performance of the framework and reducing redundant transmission. The authors in (Yao et al., 2024) maintain a database built on the edge side that stores historical requests and responses from cloud-based LLMs, effectively enhancing the knowledge richness of SLMs and improving the response quality for similar requests.

Although these works have achieved great progress in the edge-cloud model collaboration, they all operate in a static mode (Qin et al., 2024), lacking the ability to dynamically acquire and learn new information, severely restricting their adaptability and real-time learning potential. Some existing methods leverage edge devices to collect new data and upload it to the cloud, achieving dynamic knowledge integration through periodic updates of the cloud-based LLM (Fan et al., 2023; Kuang et al., 2024). However, such an update scheme is overly simple and exposes the entire architecture to several critical challenges: ~~(1) directly uploading raw data to the cloud for updating LLMs incurs substantial communication overhead;~~ (1) edge devices must wait for data transmission and cloud-side model updates, which leads to latency in user experience; (2) since new data are collected on edge devices, uploading them to the cloud for centralized updates essentially neglects the computational capacity of edge devices, reducing resource utilization; and (3) in some cases, the data may involve sensitive information, making it unsuitable for direct uploading to the cloud for training.

Offloading some computational tasks to edge devices presents a highly attractive alternative. Edge devices, such as the NVIDIA Jetson series, embedded systems with discrete GPUs or onboard intelligent computing platforms, are typically equipped with powerful CPUs, GPUs, or dedicated AI accelerators, possessing sufficient computing power to support lightweight fine-tuning of models with billions of parameters. This capability is further enhanced by Parameter-Efficient Fine-Tuning (PEFT) techniques, which significantly reduce the computational overhead required for fine-tuning with minimal impact on model performance. We compare the data transfer overhead between edge-based and cloud-based update strategy, as detailed in Table 1. The cloud-based update strategy requires two data transmissions between the edge and cloud to achieve a collaborative update (even with local storage), while the edge-based approach only needs one edge-to-cloud transmission.

To address these challenges, we propose an enhanced edge–cloud architecture that is empowered with the capability to efficiently ~~and securely~~ integrate newly collected knowledge. This improvement not only enables continuous adaptation of

Table 1: Comparison of data transfer overhead between edge-based and cloud-based update strategy.

| Method | Commonsense CQA | Math GSM8K | Natural Language WinoGrande |
|---|---|---|---|
| **Cloud-based update strategy** | | | |
| w/o Local data storage | 2.0× | 1.6× | 1.8× |
| Local data storage | 1.3× | 1.2× | 1.2× |
| **Edge-based update strategy** | **1.0×** | **1.0×** | **1.0×** |

deployed models to evolving data distributions but also ensures that knowledge updates can be incorporated with minimal latency ~~and without compromising the privacy of user data~~. In a standard edge-cloud system that serves users, cloud-based LLMs typically play an assisting and guiding role, while SLMs deployed on edge devices interact directly with users. During this process, edge device SLMs can directly access and learn from newly generated real-time data samples, thereby achieving more accurate modeling of the local data characteristics, which facilitates self-updates with minimal overhead. Driven by this intuition, we propose a novel mutual Evolution framework for edge-cloud model Collaboration called **CoEvo** that enables both cloud-side LLM and edge-side SLMs to continuously update with new knowledge. More specifically, it consists of two independent stages to update the cloud-side and edge-side models. In the cloud-to-edge stage, CoEvo incorporates confidence (Xiong et al., 2024) scores into the Chain-of-Thought (CoT) , teaching the edge-side SLMs to generate high-confidence responses that emulate the cloud-based LLM. While in the edge-to-cloud stage, the edge-side SLMs continuously acquire new domain-specific data and update themselves. CoEvo then performs credibility-based filtering on the SLMs' newly learned representations, allowing only highly reliable domain knowledge to be uploaded and used to enhance specialized inference capabilities of the cloud-side LLM. The major contributions of this paper are summarized as follows:

- We are the first to explore mutual evolution in edge-cloud model collaboration, breaking the static paradigm of traditional edge-cloud architectures and enabling an efficient ~~and secure~~ continual learning of new knowledge.

- We propose CoEvo, an enhanced edge-cloud architecture that enables perception and learning from raw data through local updates on the edge side. It also facilitates bidirectional knowledge transfer between the edge and the cloud via a credible chain-of-thought and credible probability matrices.

- We conduct extensive experiments across multiple datasets spanning various domains. Experimental results demonstrate that our method achieves performance improvements on both the edge and cloud sides compared to state-of-the-art approaches.

We employ Llama3 (Dubey et al., 2024) 8B as the edge-side SLMs and Llama3 70B as the cloud-side LLM in the edge-cloud architecture, evaluating performance across general domains (MMLU (Hendrycks et al., 2021)), commonsense reasoning tasks (CommonsenseQA(CQA) (Talmor et al., 2019)), math tasks (GSM8K (Cobbe et al., 2021)), and co-reference resolution tasks (WinoGrande (Sakaguchi et al., 2020)). In the cloud-to-edge process, CoEvo improves by 1% to 2% compared to existing baseline methods; In the edge-to-cloud process, CoEvo enables a 1% to 1.3% increase in the inference accuracy of the optimized cloud-side LLM.

## 2 RELATED WORK

**Knowledge Transfer from Cloud LLMs to Edge SLMs:** Based on edge-cloud architecture, (Xu et al., 2024; Peng et al., 2024) use cloud-based LLMs to enhance the performance of edge-side SLMs. This is achieved by building a local data store from historical interactions with the cloud LLM and dynamically integrating it with the predictions of the SLM on the device during inference (Ding et al., 2024). (Chen et al., 2024b) focuses on improving edge-side SLMs through knowledge distillation (Wang et al., 2022) while offloading all gradient-related operations to the cloud, thereby reducing the computational burden on the edge-side. (Hao et al., 2024; NING et al., 2025) leverage LLMs to provide token-level inference guidance for edge-side SLMs, integrating the LLM's semantic understanding into the actual inference process of the SLM. However, it is constrained by the upper limit of the cloud-side model's inference capabilities, and the lack of ground truth labels and chain of thought(CoT) (Wei et al., 2022b) data for given tasks (Yuan et al., 2024b) undermines its applicability. Our approach focuses on the potential of edge devices to acquire new data and aims to achieve continual learning in an edge-cloud framework by leveraging domain-specific data.

**Advanced Inference Techniques in Language Model:** The CoT-related (Wang et al., 2023b; Wan et al., 2025; Zhang et al., 2025a) technique guides language models to generate coherent thought chains and answers during inference, requiring the model to engage in one or more intermediate reasoning steps before producing the final answer (Kojima et al., 2022; Fu et al., 2023). Other studies have further enhanced the inference capabilities of language models by extending the CoT paradigm, such as integrating internal generation processes with external actions (Wang et al., 2024a) (e.g., leveraging RAG, invoking search engines, calculators, or code interpreters). They enhance the model's comprehensive inference capabilities by leveraging external knowledge, although constructing a well-structured external knowledge base or designing effective task flows is by no means an extra overhead (Cheetirala et al., 2025). The issue with inference technology is that language models themselves are constrained (Bian et al., 2024; Wang et al., 2025) by the scope of training data, delays in knowledge updates, and potential factual biases. Relying solely on inference optimization often struggles to break through the model's inherent cognitive boundaries.

**Fine-Tuning Techniques for Language Model Optimization:** Direct Preference Optimization (DPO) (Rafailov et al., 2023) eliminates the need for reward models typically required in reinforcement learning and RLHF (Reinforcement Learning from Human Feedback) (Ouyang et al., 2022) by directly incorporating preference data into the training objective. This approach reduces computational overhead while ensuring the model focuses on preferred outputs (Shankar et al., 2024). Distill-step-by-step (Hsieh et al., 2023) enhances student models' inference capabilities by aligning their outputs (both answers and rationales) with those generated by teacher models (Beyer et al., 2022). This dual alignment improves both the accuracy of problem solving and the generation of CoT. Chain of Preference Optimization (CPO) (Zhang et al., 2025b) extends DPO by incorporating multistep thought chains, where the model generates and evaluates multiple inference paths while

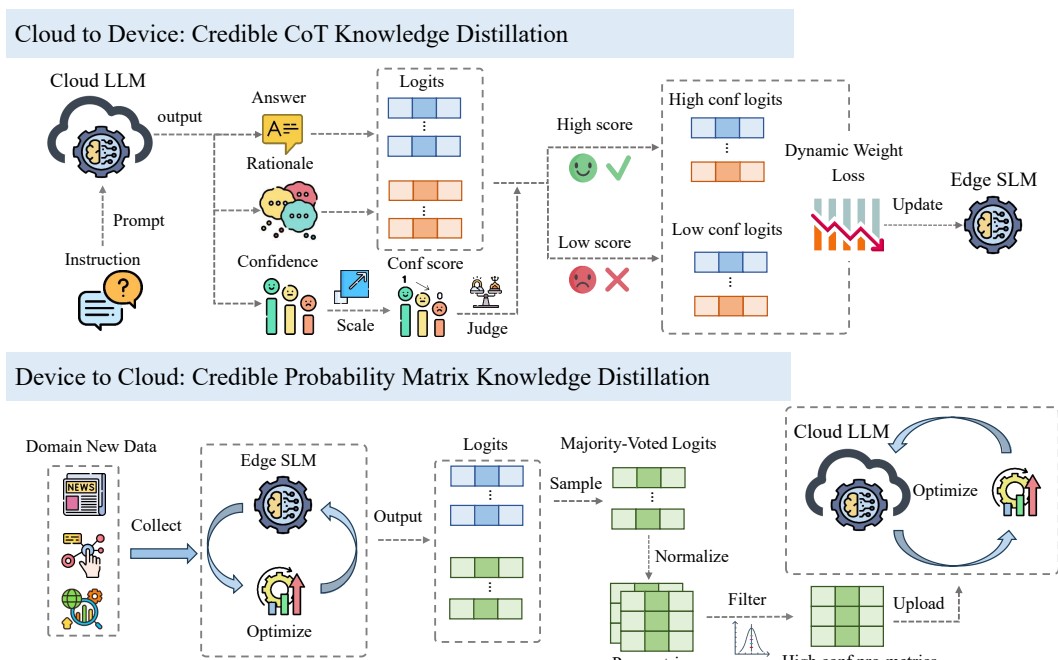

Figure 1: The overall architecture of CoEvo. In the cloud-to-edge phase, the cloud LLM performs instruction-guided inference to generate answers, rationales, and confidence scores. These outputs provide the basis for knowledge distillation, where confidence scores dynamically weight the knowledge to optimize edge SLMs. In the edge-to-cloud phase, the edge-side SLMs are optimized with new data and generate responses from historical interactions. CoEvo then applies multiple-sample voting, probability scaling, and filtering to extract high-quality domain knowledge, which is uploaded back to the cloud to further refine the cloud-side LLM.

explicitly considering dispreferred chains. A potential issue with these methods is the lack of consideration for knowledge quality, which leads to suboptimal results being incorporated into the training process. Our approach filters the data used to avoid interference from low-quality content, thereby enhancing the effectiveness of fine-tuning.

## 3 CoEvo: A Mutual Evolution Framework for Edge–Cloud Model Collaboration

### 3.1 Overview

CoEvo leverages the unique advantage of edge devices being accessible to users and extends the static edge–cloud collaboration framework into a dynamic knowledge learning paradigm, supporting the mutual evolution of models on both the edge and cloud sides. Figure 1 illustrates the details of CoEvo. In the cloud-to-edge phase, the cloud-based LLM generates rationales and labels on a general-domain dataset $D$. These outputs are distilled into a base edge-side SLM $\mathcal{M}_0$, where confidence scores are used to weight the knowledge and mitigate the impact of suboptimal outputs, resulting in an enhanced SLM $\mathcal{M}_1$ with improved semantic comprehension. $\mathcal{M}_1$ is then deployed on edge devices for domain-specific inference. In the edge-to-cloud phase, the edge-side SLMs continuously interact with the local environment and user context, serving as natural collectors of domain-specific data and enabling efficient local updates. Through this process, $\mathcal{M}_1$ is further optimized into a domain expert model $\mathcal{M}_2$. The superior domain knowledge extracted from $\mathcal{M}_2$, specifically knowledge that surpasses the cloud LLM's existing domain understanding, is selectively distilled back into the cloud-side LLM $\mathcal{M}_t$, yielding an improved model $\mathcal{M}_T$ with stronger domain inference capabilities. Through this bidirectional synergy, both edge-side and cloud-side models can co-evolve by continuously learning new knowledge and improving domain-specific performance.

## 3.2 CLOUD-TO-EDGE: CREDIBLE COT KNOWLEDGE DISTILLATION

In the cloud-to-edge stage of CoEvo, the cloud-based LLM transfers its inference capability to the edge-side SLM through knowledge distillation. Conventional methods typically rely on labels as the primary form of knowledge. However, labels alone are insufficient to enhance the semantic understanding of SLMs, limiting their ability to achieve strong domain-specific inference even after optimization. Motivated by works such as CoT distillation (Wang et al., 2023a), we instead use both labels and rationales as knowledge, thereby expanding the informational scope and enabling SLMs to learn not only outcomes but also the underlying inference processes. A critical challenge lies in the assumption that cloud-based LLMs can consistently generate labels and rationales of consistently high quality. In practice, even LLMs, despite their strong inference capabilities, may produce ambiguous or erroneous outputs, which can degrade performance. To address this issue, CoEvo introduces mechanisms that allow SLMs to emphasize high-quality knowledge while filtering out noisy or unreliable inferences, thereby maximizing the effectiveness of the distillation process.

Confidence is commonly used to reflect the degree of self-assurance that large models have in their outputs, which may correlate with the correctness of the inference results. We analyze the confidence generated by LLM inference across multiple datasets from different domains, as shown in Figure 2. Two key observations emerge: (1) LLMs generally exhibit high confidence in their responses (consistently above 0.67), regardless of correctness, which aligns with findings in prior studies; and (2) despite this overall tendency toward high confidence, correct answers are still associated with significantly higher confidence than incorrect ones. These observations suggest a potential correlation between confidence levels and the quality of the responses. In particular, the validity of the inference process

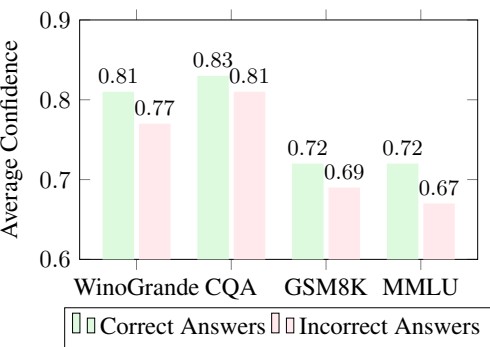

Figure 2: Comparison of average confidence scores for correct vs. incorrect answers across different groups.

(i.e., the generated CoT) is strongly tied to the correctness of the final answer: a coherent rationale typically leads to a correct result. Thus, confidence can serve as a useful indicator of both rationale quality and answer reliability. Nonetheless, we also observe cases where LLMs produce correct answers with low confidence, reflecting a lack of self-assurance. In such cases, multiple sampling often increases the likelihood of generating incorrect responses. Therefore, it is advisable to assign lower weights to these low-confidence samples during training, even if their answers are occasionally correct, to prevent the student model from inheriting similar confusion.

We let the cloud-based LLM $\mathcal{M}_t$ execute inference tasks on a problem set $\mathcal{D}$. $\mathcal{M}_t$ generates corresponding rationale ($\mathcal{Q} \to \mathcal{R}$), answers ($\mathcal{Q} \to \mathcal{A}$), and confidence ($\mathcal{Q} \to \mathcal{C}$) for each instruction. We perform knowledge distillation on the SLM. Given the preceding sequence $(1, \ldots, i-1)$, $\mathcal{M}_t$ generates a prediction for the $i^{th}$ token. Based on the answers and rationales obtained, the fundamental objectives are formulated as follows:

$$\mathcal{L}_\mathcal{A} = -\log p([\mathcal{A}_i \mid \mathcal{P}_{Q \to A}; Q; A_{<i}]; \mathcal{M}_t). \tag{1}$$

$$\mathcal{L}_\mathcal{R} = -\log p([\mathcal{R}_i \mid \mathcal{P}_{Q \to R}; Q; R_{<i}]; \mathcal{M}_t). \tag{2}$$

where $p$ represents probability, $\mathcal{A}_i$ and $\mathcal{R}_i$ depict the $i^{th}$ token in the answer and rationale, $< i$ refers to the sequence of tokens preceding the $i^{th}$ token, $[;]$ denotes the operation that formats the LLM input by assembling question, answer, and rationale into the prompt $\mathcal{P}_t$. We scale the training objectives in (1) and (2) using the confidence:

$$\mathcal{L} = S_{conf} \cdot [(1-\alpha)\mathcal{L}_R + \alpha\mathcal{L}_A]. \tag{3}$$

where $\alpha$ is hyperparameter, $S_{conf}$ represents the confidence score in the model inference. By modeling knowledge distillation as the process above, we improve SLM $\mathcal{M}_0$ to obtain $\mathcal{M}_1$. $\mathcal{M}_1$ is deployed on an edge device to perform practical tasks. Not only does $\mathcal{M}_1$ surpass $\mathcal{M}_0$ in inference ability, but it also possesses stronger cognitive understanding abilities. This makes it possible for $\mathcal{M}_1$ to evolve into a domain-specific expert after being exposed to domain-specific data.

### 3.3 EDGE TO CLOUD: CREDIBLE PROBABILITY MATRIX KNOWLEDGE DISTILLATION

CoEvo primarily leverages edge-side SLMs to acquire new data. By learning from evolving data distributions and performing local self-updates, the SLMs dynamically adapt to new knowledge. These locally acquired updates are then fed back to the cloud, where they further enhance the performance of the LLM. We first leverage the newly acquired data to optimize SLM $\mathcal{M}_1$ on the edge device, resulting in a domain-specific expert $\mathcal{M}_2$. While inheriting $\mathcal{M}_1$'s strong inference abilities, $\mathcal{M}_2$ also possesses richer domain-specific knowledge. We select a portion of the instructions used during the self-updating process and combine them with $\mathcal{M}_2$'s responses to these instructions as knowledge to be uploaded to the cloud. To obtain superior responses, for each instruction, CoEvo samples multiple inference results $y$ and selects the one with the highest consistency as the interaction record $y^*$ by majority voting. Through inference sampling, CoEvo aims to obtain responses that approach the upper limit of $\mathcal{M}_2$'s inference capability.

$$y^* = \arg \max_{y \in \mathcal{Y}} \sum_{i=1}^{N} \mathbb{I}(y_i = y). \tag{4}$$

We selectively extract interaction records in order to identify and upload the knowledge where $\mathcal{M}_2$ produced relatively high-quality responses to the cloud ~~while minimizing the impact of transmitting raw responses on original data privacy~~. For each response of $\mathcal{M}_2$, we obtain the probability matrix $P$ of its response component:

$$\mathbf{P} = \begin{pmatrix} p_{11} & p_{12} & \cdots & p_{1m} \\ p_{21} & p_{22} & \cdots & p_{2m} \\ \vdots & \vdots & \ddots & \vdots \\ p_{n1} & p_{n2} & \cdots & p_{nm} \end{pmatrix}, \quad p_{ij} = \frac{e^{z_{ij}}}{\sum_{k=1}^{m} e^{z_{ik}}}. \tag{5}$$

where $m$ is the length of the response sequence, $z$ represents the logits corresponding to the $i^{th}$ token in the response sequence, and $n$ is the size of the vocabulary. Due to the large value of $n$(often tens or hundreds of thousands), computing and transmitting the probability matrix for the entire sequence becomes computationally expensive. To address this, CoEvo therefore applies a compression process. Through statistical analysis across the multiple datasets used in our experiments, we find that the probability distribution is highly concentrated in the top-k elements. We select the top 10 elements, which significantly reduces the raw transmission cost while striving to preserve the uncertainty information inherent in the model's predictions. To obtain superior responses, we reconsider the relationship between confidence and inference outcomes. Confidence in a language model's response is directly reflected in the magnitude of its larger output logits. This same principle applies to evaluating the reliability of a probability matrix. For a given interaction record with a probability matrix, CoEvo calculates: (1) the sum of the top-k probabilities and (2) the maximum probability in the top-k probabilities.

$$p_{t1} = \begin{cases} 1, & \text{if } \sum_{i=1}^{k} P_{ij} > p_1 \quad \forall j \in \{1, \dots, m\}, \\ 0, & \text{otherwise.} \end{cases}, p_{t2} = \begin{cases} 1, & \text{if } P_{1j} > p_2 \quad \forall j \in \{1, \dots, m\}, \\ 0, & \text{otherwise.} \end{cases}$$

$$\tag{6}$$

If $p_{t1}$=1 and $p_{t2}$=1, the interaction record is considered high quality:

$$p_{final} = \begin{cases} 1, & \text{if } p_{t1} = 1 \land p_{t2} = 1, \\ 0, & \text{otherwise.} \end{cases} \tag{7}$$

We upload records with $p_{final}$=1 to the cloud, feeding high-quality domain knowledge back to the cloud-side LLM. Through credible probability matrix knowledge distillation, we optimize the LLM $\mathcal{M}_t$ to obtain $\mathcal{M}_T$. $\mathcal{M}_T$ breaks through the original performance ceiling of the cloud-side LLM by achieving dynamic updates based on edge-side new knowledge and serves as the foundational cloud-side LLM for the next iteration of CoEvo.

## 4 EXPERIMENTS

### 4.1 EXPERIMENT SETUP

**Datasets.** We evaluate CoEvo on four datasets from diverse domains: (1) Multi-task understanding: **MMLU** (Hendrycks et al., 2021); (2) Commonsense inference: **CQA** (Talmor et al., 2019); (3)

Table 2: Performance comparison between CoEvo and baselines in the cloud-to-edge stage

| Method | Multi-task MMLU | Commonsense CQA | Math GSM8K | Natural Language WinoGrande |
|---|---|---|---|---|
| Meta Llama3 8B | 57.26 | 59.38 | 52.15 | 62.46 |
| **Efficient inference based on CoT** | | | | |
| Chain of Thought | 59.48 | 63.68 | 59.22 | 62.77 |
| Self-Consistency CoT | 61.17 | 65.00 | 60.09 | 62.98 |
| Tree of Thought | 59.95 | 64.64 | 58.73 | 62.48 |
| **Fine-tuning-based optimization** | | | | |
| DPO | 63.90 | 64.06 | 61.66 | 67.10 |
| Distill step by step | 64.38 | 65.44 | 59.31 | 68.20 |
| SPIN | 63.85 | 63.19 | 62.08 | 66.62 |
| CoEvo (Ours) | **65.91** | **66.26** | **62.44** | **70.40** |

**Math problems**: GSM8K (Cobbe et al., 2021); (4) Natural Language Inference: **WinoGrande** (Sakaguchi et al., 2020). Details of the datasets can be found in Appendix A.

**Baseline.** We separately evaluate the cloud-to-edge and edge-to-cloud processes to validate the performance improvements of both the edge-side SLM and cloud-side LLM. For the cloud-to-edge process, we compare two types of approaches. (1) Efficient inference methods based on CoT: **Chain of Thought** (Wei et al., 2022b), **Self-Consistency CoT** (Wang et al., 2023b), and **Tree of Thought** (Yao et al., 2023); (2) Fine-tuning-based optimization methods: **DPO** (Rafailov et al., 2023), **Distill Step-by-Step** (Hsieh et al., 2023), and **SPIN** (Chen et al., 2024c). For the edge-to-cloud process, we compare the results with both the base LLM and the LLM after Supervised Fine-Tuning (SFT) (Wei et al., 2022a). Details of the baselines can be found in Appendix B.

**Implementation Details.** We use Llama3-8B (Dubey et al., 2024) as the edge-side SLM and Llama3-70B as the cloud-side LLM in our experiments. On the edge side, we limit our experiments to 1 A100 80GB GPU(under 42 GB VRAM used in total) to simulate limited computational resources; On the cloud side, we employ a batch size of 32 or 64 using 16 A100 80GB GPUs. The training phase employs the AdamW optimizer with cosine annealing and 20 warmup steps. We employ machine learning libraries such as Deepspeed (Rajbhandari et al., 2020) to facilitate model training and utilize the LoRA technique to reduce the resource overhead required for edge-based training. Details of configurations can be found in Appendix C.

## 4.2 Performance Overview

Table 2 presents a comparison between CoEvo and baseline methods in cloud-to-edge stage. CoEvo consistently outperforms most baseline methods across the four task domains. On MMLU, CoEvo achieves a gain of 1.5% over the strongest baseline, demonstrating its effectiveness

Table 3: Performance comparison in edge-to-cloud stage.

| Method | Commonsense CQA | Math GSM8K | Natural Language WinoGrande |
|---|---|---|---|
| Meta Llama3 70B | 78.09 | 80.20 | 82.42 |
| SFT | 79.09 | **81.74** | **83.97** |
| **CoEvo(Ours)** | **79.26** | 81.18 | 83.79 |

in cross-domain tasks. On CQA, it improves by 0.8% compared to the best baseline, achieving a marginal gain. Our analysis reveals that the answers generated by cloud-side LLM in cloud-to-edge stage were actually suboptimal, since the use of CoT-based inference on CQA reduced the accuracy of the inference. This may be attributed to the commonsense nature of CQA, for which the CoT method is less suitable. On WinoGrande, CoEvo outperforms the best baseline by approximately 2%, demonstrating the effectiveness of our method in enhancing the SLM's semantic understanding ability. On GSM8K, CoEvo exhibits slightly higher~~lower~~ performance compared to the best baseline. ~~A potential explanation is the fact that in the mathematical domain, the evaluation of CoT quality is not fixed. For example, multiple solution approaches can lead to correct results, and this characteristic may have interfered with CoEvo's confidence-based evaluation process.~~ Fur-

Table 4: Performance analysis of different design choices.

| Ablation ID | Confidence Evaluation | Inference Sampling | Response Filtering | Commonsense CQA | Math GSM8K | Natural Language WinoGrande |
|---|---|---|---|---|---|---|
| 1 | Yes | No | No | 78.64 | 79.19 | 81.13 |
| 2 | No | Yes | No | 78.71 | 79.41 | 81.34 |
| 3 | No | No | Yes | 78.02 | 79.01 | 81.47 |
| 4 | Yes | Yes | No | 78.71 | 80.76 | 83.72 |
| 5 | Yes | No | Yes | 78.84 | 80.44 | 83.50 |
| **CoEvo** | Yes | Yes | Yes | **79.26** | **81.18** | **83.79** |

thermore, while CoT-based baseline methods lead to a multiplicative increase in inference latency, CoEvo requires only single step inference to generate responses, thereby preserving the low-latency requirement of the edge-side SLM in edge-cloud architecture. In contrast to methods like DPO that require the parallel training of both a base and a policy model, CoEvo's single-model training mode in cloud-to-edge stage significantly reduces computational overhead and simplifies the training pipeline. More discussions and results on model performance and communication efficiency are available in Appendix D.

Table 3 compares the performance of CoEvo in the edge-to-cloud stage with the base model and the SFT-tuned LLM. CoEvo achieves a gain of 1.1% to 1.3% over the base model on CQA and WinoGrande, demonstrating its effectiveness in leveraging SLM to inversely optimize the LLM within the edge-to-cloud framework. On GSM8K, CoEvo improves by nearly 1% compared to the original model, indicating that CoEvo remains effective for mathematical tasks. Moreover, the performance of CoEvo is comparable to that of the LLM optimized with SFT on all datasets. This demonstrates the robust adaptability of the self-updating SLM at the edge to novel data distributions and the high quality of the knowledge uploaded to the cloud. CoEvo enables edge-to-cloud model feedback optimization in most domains and exhibits promising potential for continual learning.

## 4.3 Ablation Study

We analyze the effectiveness of each subdesign in CoEvo in different stages, as shown in Table 4. We evaluate the following three settings: (1) Whether to use confidence in cloud-to-edge stage; (2) Whether to perform multiple sampling during response generation in edge-to-cloud stage; and (3) Whether to filter candidate responses in edge-to-cloud stage. The results indicate that incorporating confidence consistently outperforms using only label and rationale data. Furthermore, performing multiple sampling during response generation in the edge-to-cloud stage can enhance the diversity and quality of output. Meanwhile, filtering candidate responses generally leads to better performance than using all responses. These findings suggest that every component of CoEvo is indispensable and effective, collectively contributing to its robust performance across diverse scenarios.

Furthermore, we conduct a detailed evaluation of each component of CoEvo to explore potential optimal configurations. Specifically, we examine: (1) the impact of different confidence weight coefficients on the optimization of edge-side SLMs; (2) the effect of varying sampling counts on the feedback performance of the cloud-side LLM; and (3) the influence of different filtering strategies on the cloud feedback process. The results are presented in Figure 3. We observe that reducing the confidence weight degrades performance, confirming the effectiveness of CoEvo confidence acquisition and scaling strategy. Increasing the sampling count generally improves inference performance, though with diminishing marginal returns, which highlights the need to balance computational cost against performance gains. Finally, applying $p_{t1}$ or $p_{t2}$ individually for data filtering yields consistently inferior results compared to their joint use. This validates CoEvo's dual-threshold filtering strategy, which considers both the highest and relatively high confidence values, thereby enhancing robustness beyond what a single-threshold approach can achieve.

As mentioned in Section 3.3, during the edge-to-cloud phase, CoEvo utilizes a credible probability matrix as the primary knowledge to optimize the LLM, which differs from the commonly used label-based fine-tuning strategies. We analyze the impact of CoEvo, conventional methods, and

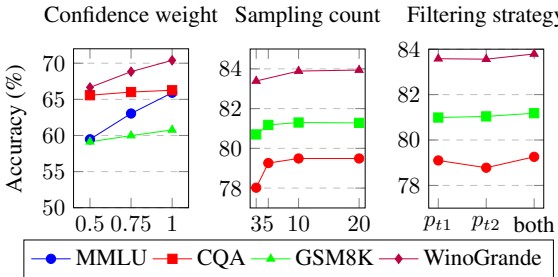 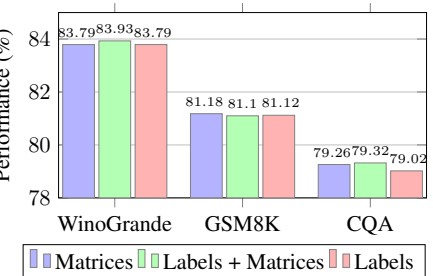

Figure 3: Additional ablation studies on the confidence weight (cloud-to-edge stage), sampling count, and filtering strategies (edge-to-cloud stage).

Figure 4: Performance comparison of optimization variants (matrices, labels, and hybrid) in edge-to-cloud stage.

Table 5: Evaluation between SFT-based Model and CoEvo in the edge-to-cloud stage.

| Methods | MMLU | CQA | GSM8K | WinoGrande |
|---|---|---|---|---|
| Meta Llama3 70B | 73.24 | 78.09 | 80.20 | 82.42 |
| **SFT-based** | | | | |
| $\mathcal{M}_{-mmlu}$ | 75.29 | 78.02 | 80.18 | **83.39** |
| $\mathcal{M}_{-cqa}$ | 72.08 | 79.09 | 80.22 | 83.43 |
| $\mathcal{M}_{-GSM8K}$ | 73.47 | 77.97 | 81.74 | 81.32 |
| $\mathcal{M}_{-wino}$ | 72.44 | 78.08 | 79.48 | 83.97 |
| **CoEvo** | | | | |
| $\mathcal{M}_{T-cqa}$ | 73.18 | 79.26 | **80.33** | 83.09 |
| $\mathcal{M}_{T-GSM8K}$ | **74.15** | 78.00 | 81.18 | 82.11 |
| $\mathcal{M}_{T-wino}$ | 72.89 | **78.14** | 79.80 | 83.79 |

their combinations on the LLM's inference performance. Specific results are shown in Figure 4. We observe that using labels, or even a combination of labels and matrices, yields almost comparable or even slightly inferior optimization results for the cloud-based LLM compared to using matrices only. CoEvo, in contrast, utilizes credible probability matrices as knowledge to optimize the cloud-based LLM. This approach excludes the interference from suboptimal outcomes, thereby ensuring both the effectiveness and efficiency of the LLM update process.

## 4.4 WHY NOT UPLOAD DOMAIN-SPECIFIC DATA DIRECTLY TO CLOUD?

As mentioned in Section 3.3, edge devices can upload new data directly to the cloud to improve LLM. However, the issue is that ~~directly uploading raw data contradicts the sensitivity to data privacy inherent in the edge-cloud distributed architecture. CoEvo uses response content based on a probability matrix, which partially obscures data details. Furthermore,~~ directly fine-tuning the LLM with in-domain data may compromise its original "generalist" nature. As a continually learning edge-cloud architecture, CoEvo must ensure that the general performance of the cloud-side LLM remains consistently high. By using cleaned and filtered probability matrices as fine-tuning data, CoEvo indirectly provides high-quality domain new data while mitigating catastrophic forgetting and minimizing the impact on the LLM's fundamental inference capabilities. We evaluate the inference performance of the optimized LLM obtained by different methods in other domains.

As shown in Table 5, although the domain-specific inference capabilities of the LLM fine-tuned via SFT on MMLU remain largely unaffected ($\mathcal{M}_{-mmlu}$), LLMs fine-tuned on other domain-specific datasets exhibit varying degrees of degradation in cross-domain inference performance. Leveraging the knowledge matrix from the feedback of SLM instead of hard labels, CoEvo reduces the interference caused by single domain data on the overall inference ability of the LLM while ensuring stable performance improvement in the target domain.

Table 6: Comparison of computational overhead between CoEvo and simpler strategy.

| Method | Local optimization | Multiple-sample voting | Probability matrices filtering | Total |
|---|---|---|---|---|
| transmitting data only | 0 | 0 | 0 | 0 |
| update + transmitting data | 3.9h | 5mins | 0 | 3.98h |
| **CoEvo** | 3.9h | 26mins | 1s | 4.32h |

Another factor is the combined cost of computational resources and data transmission in the edge-cloud paradigm. Compared to directly uploading data to the cloud, the local optimization and majority voting mechanisms performed on the edge side in CoEvo introduce a certain amount of computational overhead. We analyze the computational overhead of CoEvo in comparison to both directly uploading data to the cloud (transmitting data only) and the method of transmitting data after local updates (update + transmitting data). The specific results are shown in Table 6.

"Update + transmitting data" corresponds to the sample with Ablation ID 1 in Table 4. As can be seen, it not only fails to significantly reduce the computational overhead on the edge side but also considerably degrades the performance of cloud-side optimization. Regarding "transmitting data only", this essentially represents the traditional cloud-based update approach, which means it necessitates the additional transmission of the cloud-based LLM's response data to update the edge-side SLM. Detailed results are shown in Table 1.

~~We compare the data transfer overhead between CoEvo and cloud-based update strategy, as detailed in Table 1. The cloud-based update strategy requires two data transmissions between the edge and cloud to achieve a collaborative update. Even with local storage of new data, it still necessitates the additional transmission of the cloud-based LLM's response data to update the edge-side SLM. In contrast, CoEvo completes the SLM update directly on the edge side. It only needs to transmit a portion of the new data and its own responses to the cloud to complete the update of models on both sides, thereby saving the additional overhead of data transmission.~~

## 5 CONCLUSION

We present CoEvo, a trainable framework designed to break static mode of edge-cloud architecture and enhance the performance of language models by applying credible CoT knowledge distillation and credible probability matrix knowledge distillation. CoEvo demonstrates the characteristic of proximity to edge devices and users, leveraging the ability to access new domain-specific data during interaction to enable dynamic knowledge flow in the edge-cloud architecture. Through cloud-to-edge model optimization, CoEvo produces a strengthened SLM with robust inference capabilities; through edge-to-cloud feedback, it yields an LLM with improved performance in specific domains. Furthermore, CoEvo achieves continual evolution through multiple self-iterations, extending its benefits across more domains.

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

# APPENDIX

## THE USE OF LARGE LANGUAGE MODELS (LLMS)

In preparing this work, we have used Large Language Models (LLMs) exclusively for the purposes of translation and language polishing. The content, arguments, and conclusions presented herein are entirely my own, and the use of LLMs did not contribute to the generation of original ideas or substantive content.

## ETHICS STATEMENT

This work complies with the ICLR Code of Ethics. No human subjects, personally identifiable data, or sensitive datasets were involved. The use of Large Language Models (LLMs) was strictly limited to translation and language polishing; they did not contribute to the generation of original ideas, methodology, results, or conclusions.

## REPRODUCIBILITY STATEMENT

We have taken steps to ensure the reproducibility of our work. The experimental setup, including datasets, preprocessing steps, and hyperparameters, is detailed in Section 4 and the Appendix. Source code and instructions for reproducing our results are available if needed.

## A  DATASETS

**MMLU:** A large-scale, widely used multi-task dataset that covers 57 distinct subjects, designed to evaluate a model's breadth of knowledge and inference abilities across various disciplinary domains. It consists of a development set, a validation set, and a test set, comprising approximately 14,000 questions in total.

**CommonsenseQA(CQA):** A dataset designed to evaluate models' commonsense inference capabilities. After partitioning, it contains 9,741 training instances and 1,221 test instances.

**GSM8K:** A dataset specifically designed to evaluate models' mathematical inference capabilities, consisting of grade-school math word problems that require multiple logical steps to solve. It includes a training set of 7,473 samples and a test set of 1,319 samples.

**WinoGrande:** A large-scale natural language inference dataset, primarily used to evaluate the ability to resolve coreference resolution tasks. It consists of a training set with 40,398 samples, a validation set with 1,267 samples, and a test set with 1,767 samples.

Table 7 in the Appendix shows sample examples from each dataset.

## B  BASELINES

**Efficient inference based on CoT:**

- **CoT:** It is an algorithm that guides LLMs to decompose "question-answer" into "question-thinking-answer" through "think step by step" prompting. It has been widely adopted in the field of LLM research.

- **self-consistency CoT:** It is an algorithm that extends CoT. This method enhances the robustness and accuracy of CoT by generating multiple distinct inference paths and selecting the most consistent answer from these paths through a majority voting mechanism.

- **ToT:** This is an algorithm that integrates tree search concepts into the inference process of large language models. It transforms the traditional single-path expansion into a tree-structured exploration framework, with the key idea being that the language model can evaluate multiple intermediate steps during inference through methods such as voting or scoring. These intermediate steps correspond to nodes in the tree, and the inference path is

Table 7: Sample examples from different datasets belonging to diverse task domains.

| Dataset | Instructions and GT Answers for Each Dataset used to evaluate CoEvo |
|---------|---------------------------------------------------------------------|
| MMLU | **Question:** Which statement best explains the purpose of Hart's distinction between 'being obliged' and 'having an obligation'? 
 **Answer:** It illuminates the concept of a rule. |
| CQA | **Question:** A John is a bum. Much like the stereotype, he lives near this sort of transportation infrastructure. Where does he live? 
 **Answer:** bridge. |
| GSM8K | **Question:** The ratio representing the age of Halima, Beckham, and Michelle is 4:3:7, respectively. If the total age for the three siblings is 126, calculate the age difference between Halima and Beckham. 
 **Answer:** 9. |
| WinoGrande | **Question:** The ring that I bought my girlfriend was worse than the bracelet that I bought her because the ___ was more expensive. 
 **Answer:** bracelet. |

Table 8: Experiment Setup. Dataset configurations and parameter settings.

| Attributes | cloud-to-edge stage | | | | edge-to-cloud stage | | |
|------------|-------|-----|-------|------------|-----|-------|------------|
|            | **MMLU** | **CQA** | **GSM8K** | **WinoGrande** | **CQA** | **GSM8K** | **WinoGrande** |
| Task size | 6MB | 3.7MB | 4.9MB | 8.2MB | 3.7MB | 4.9MB | 8.2MB |
| Instruction number | 13985 | 10962 | 8792 | 43432 | 10962 | 8792 | 43432 |
| Task Scenario | Multi-task | Commonsense | Math | Natural Language | Commonsense | Math | Natural Language |
| Batch Size | $s = 64$ | $s = 32$ | $s = 32$ | 32 | $s = 32$ | $s = 32$ | $s = 32$ |
| Learning Rate | $l = 1e\text{-}5$ | $l = 1e\text{-}5$ | $l = 1e\text{-}5$ | $l = 2e\text{-}5$ | $l = 1e\text{-}5$ | $l = 1e\text{-}5$ | $l = 2e\text{-}5$ |
| Local training epoch | $E = 5$ | $E = 5$ | $E = 3$ | $E = 3$ | $E = 5$ | $E = 3$ | $E = 3$ |

dynamically optimized through backtracking or heuristic search methods (such as breadth-first search or depth-first search).

**Fine-tuning-based optimization methods:**

- **DPO:** It is an algorithm that fine-tunes language models by directly optimizing human preference data. It bypasses the complex reward model training steps of traditional RLHF by re-formulating the preference learning problem as a policy-based loss function, which directly maximizes the logarithmic probability difference between preferred and non-preferred responses through a simple binary cross-entropy objective.

- **Distill step by step:** This algorithm first trains a small model to generate intermediate inference steps (rationales), then jointly optimizes them with the final answer prediction. This approach enables the model to learn and acquire the logical inference capabilities of larger models while maintaining its lightweight nature.

- **SPIN:** This algorithm builds upon DPO and enables continual optimization of large models through a self-play mechanism, without relying on additional human preference data or reinforcement learning frameworks. The key lies in modeling the fine-tuning process as a "two-player game" task: the main model (current version) and a historical model (previous version) generate responses to questions, and a contrastive learning objective (similar to DPO) drives the main model to progressively surpass the inference performance of the historical model.

## C CONFIGURATIONS

We document the specific settings and implementation details of all experiments mentioned in the main text to ensure the reproducibility of the research. When replicating various baseline methods, we strictly adhere to the design principles outlined in their original papers and implement them based on publicly available code repositories.

Table 9: Performance of CoEvo and baseline methods on other heterogeneous model pairs.

| Methods | cloud-to-edge stage | edge-to-cloud stage |
|---|---|---|
| | MMLU | CQA |
| Qwen3 1.7B | 56.97 | – |
| Qwen3 14B | – | 81.80 |
| Distill step by step | 62.90 | – |
| SPIN | 63.12 | – |
| CoEvo | **64.72** | **83.04** |

For efficient inference methods based on CoT: (1) CoT: Standard chain-of-thought inference is triggered using a few-shot prompting. (2) Self-Consistency CoT: For each question, balancing inference overhead and sampling diversity, the model generates five inference paths and selects the most consistent answer through a "majority voting" mechanism. (3) ToT: At each inference step, the model generates three candidate inference paths. Depending on the specific task, the depth of the inference tree (i.e., the maximum number of inference steps) is set between two and three. At each step, three evaluations are performed to filter the optimal inference path.

For fine-tuning-based optimization methods: (1) DPO: We use the responses generated by cloud-side LLM as preferred data and randomly select one from the remaining answer options as non-preferred data. (2) Distill Step by Step: The original settings are followed. (3) SPIN: We use the responses generated by cloud-side LLM as the initial SFT data.

More experimental details can be found in Table 8 in the Appendix.

## D  ADDITIONAL RESULTS

### D.1  EXPERIMENTAL VALIDATION ON OTHER HETEROGENEOUS MODEL PAIRS

The effectiveness of the CoEvo method is validated on heterogeneous model pairs (Qwen1.7B/14B), with specific results shown in Table 9. Our new experimental results consistently demonstrate that our framework remains effective on the datasets evaluated using the Qwen model pair. This positive outcome strongly supports the universality of our method across different model architectures and scale ratios.

### D.2  COMPARATIVE ANALYSIS OF TEXT PERPLEXITY

Perplexity (PPL) is a fundamental metric for evaluating the performance of language models by measuring their uncertainty in predicting text sequences. Given a sequence of tokens $W = (w_1, w_2, \ldots, w_N)$, perplexity is computed as:

$$\text{PPL}(W) = \exp\left(-\frac{1}{N}\sum_{i=1}^{N}\log P(w_i|w_1,\ldots,w_{i-1})\right). \tag{8}$$

where:

- $N$: Total number of tokens in the evaluation sequence
- $w_i$: The $i$-th token in the sequence
- $P(w_i|w_1,\ldots,w_{i-1})$: Conditional probability of token $w_i$ given preceding context
- $\log$: Natural logarithm (base $e$)

Lower perplexity values indicate better model performance, with a theoretical minimum of 1 (perfect prediction) and a baseline value equal to the vocabulary size for random guessing. This metric

Table 10: Text perplexity comparison of proposed and baseline methods.

| Methods | cloud-to-edge stage | | | | edge-to-cloud stage | | |
|---|---|---|---|---|---|---|---|
| | MMLU | CQA | GSM8K | WinoGrande | CQA | GSM8K | WinoGrande |
| Meta Llama3 8B | 4.49 | 10.26 | 6.77 | 3.47 | – | – | – |
| Meta Llama3 70B | – | – | – | – | 7.91 | 8.43 | 2.27 |
| **Efficient inference based on CoT** | | | | | | | |
| Chain of Thought | 4.08 | 9.44 | 6.56 | 3.29 | – | – | – |
| Self-Consistency CoT | 4.08 | 9.44 | 6.56 | 3.29 | – | – | – |
| Tree of Thought | 3.40 | 8.08 | 5.94 | 3.01 | – | – | – |
| **Fine-tuning-based optimization** | | | | | | | |
| DPO | 3.37 | 7.43 | 5.67 | 2.87 | – | – | – |
| Distill step by step | 3.25 | 6.82 | 5.98 | 3.06 | – | – | – |
| SPIN | 3.28 | 7.52 | 5.91 | 3.03 | – | – | – |
| CoEvo | **2.90** | **6.51** | **5.60** | **2.59** | **7.65** | **7.99** | **2.22** |

reflects how "surprised" the model is when encountering the test data, with well-calibrated models achieving lower perplexity on in-distribution text.

Following the experimental setup outlined in Sections 4.1 and 4.2, we evaluate the perplexity of responses generated by CoEvo and baseline approaches across multiple datasets. The test results are summarized in Table 10 in the Appendix.

During cloud-to-edge and edge-to-cloud phase, the perplexity achieved by CoEvo outperforms existing baseline methods on most datasets. Although superior perplexity does not directly equate to correct inference, it reflects the model's clarity in understanding problems and has become one of the widely adopted evaluation metrics in the field of LLMs.

## D.3 THE IMPACT OF DIFFERENT OPTIMIZATION OBJECTIVES ON FINE-TUNING OUTCOMES

As mentioned in Section 3.3, we design a joint optimization objective based on three key metrics: answer, rationale, and confidence. Then facilitate the distillation of knowledge from the LLM on the cloud to the SLM on the edge. In this section, we elaborate on the exploration of specific formulations for this joint optimization objective during the experimental phase. We designed three variants of the objective function:

$$\mathcal{L} = conf \cdot (1 - \alpha)\mathcal{L}_{\mathcal{R}} + \alpha\mathcal{L}_{\mathcal{A}}. \tag{9}$$

where $\alpha$ is a hyperparameter. The design intuition is to use the original confidence score to adjust the final answer. When the confidence is low, the weight of the final answer in the training objective is reduced, guiding the model to focus more on learning the inference process.

$$\mathcal{L} = (1 - conf)\mathcal{L}_{\mathcal{R}} + conf \cdot \mathcal{L}_{\mathcal{A}}. \tag{10}$$

The idea here is to dynamically balance the contributions of $\mathcal{L}_R$ and $\mathcal{L}_A$ using the confidence score. When confidence is high, the training objective shifts emphasis toward the rationale.

$$\mathcal{L} = \exp(conf) \cdot [(1 - \alpha)\mathcal{L}_{\mathcal{R}} + \alpha\mathcal{L}_{\mathcal{A}}]. \tag{11}$$

where $\alpha$ is a hyperparameter. This formulation uses the exponentially scaled confidence score to dynamically adjust the weighted sum of $\mathcal{L}_R$ and $\mathcal{L}_A$.

We evaluate the performance of the optimized SLM with these three variants in various datasets. We denote the SLM obtained by applying Eq. 9 as $\mathcal{M}_{1-A}$, the SLM obtained by applying Eq. 10 as $\mathcal{M}_{1-B}$, and the SLM obtained by applying Eq. 11 as $\mathcal{M}_{1-C}$. We selected distill step by

Table 11: Performance of SLM Variants Optimized with Different Objective Functions.

| Methods | MMLU | CQA | GSM8K | WinoGrande |
|---------|------|-----|-------|------------|
| Meta Llama3 8B | 57.26 | 59.38 | 52.15 | 62.46 |
| Distill step by step | 64.38 | 65.44 | 59.31 | 68.82 |
| $\mathcal{M}_{1-A}$ | **65.91** | **66.26** | **60.77** | 68.71 |
| $\mathcal{M}_{1-B}$ | 64.28 | 64.79 | 59.20 | 67.88 |
| $\mathcal{M}_{1-C}$ | 63.60 | 63.25 | 57.11 | **70.40** |

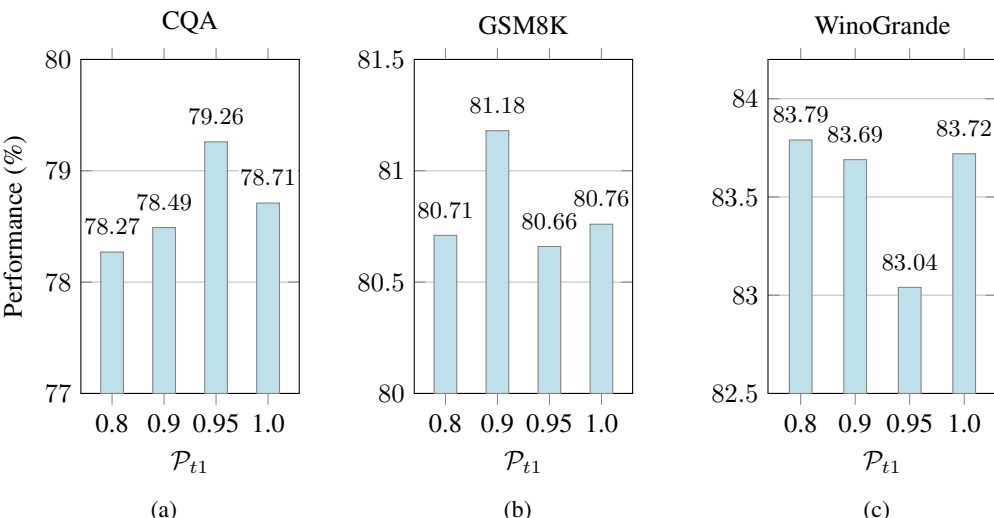

Figure 5: Performance comparison under different $\mathcal{P}_{t1}$ thresholds on three datasets. Given the presentation requirements, $\mathcal{P}_{t1}$=1 here actually represents using the complete dataset.

step—which also employs a composite optimization objective utilizing both labels and rationales, but does not incorporate confidence—as the baseline for comparison. The results are shown in Table 11.

$\mathcal{M}_{1-A}$ achieved the best performance on the three datasets: MMLU, CQA and GSM8K, while $\mathcal{M}_{1-C}$ performed optimally on WinoGrande. This indicates that it is meaningful to analyze and construct different forms of optimization objectives tailored to tasks across various domains. It should also be noted that the construction method $\mathcal{M}_{1-B}$ doesn't achieve satisfactory results on any of the test datasets, indicating that improper utilization of the confidence metric may disrupt its intrinsic relationship with labels and rationales, thereby leading to suboptimal outcomes.

### D.4 THE EFFECT OF DATA FILTERING THRESHOLDS ON CLOUD FEEDBACK PERFORMANCE

In this section, we supplement the exploration of a key sub-technique in edge-to-cloud stage of Co-Evo: response filtering. The objective of response filtering is to obtain high quality knowledge from $\mathcal{M}_2$ - the 'domain-specific expert' enhanced in cloud-to-edge stage and enhancement of domain-specific data. Specifically, for responses generated by $\mathcal{M}_2$ based on historical interactions, CoEvo filters qualified high-quality answers according to the design described in Section 3.3, and uploads these to the cloud as raw knowledge for feedback optimization of the cloud-side LLM. We explored the effect of $\mathcal{P}_{t1}$—that is, the ratio of the maximum probability to $\mathcal{P}_{t1}$—under different thresholds on the optimization of the cloud-side LLM. Experimental results are shown in Figure 5 in the Appendix.

We sequentially set $\mathcal{P}_{t1}$ to 0.8, 0.9, 0.95, and 1.0. It should be noted that, for ease of presentation, $\mathcal{P}_{t1}$=1.0 actually represents using the complete dataset without any filtering. The setting of $\mathcal{P}_{t1}$ exhibited varying impacts on the optimization of the cloud-side LLM across different domain datasets. For CQA, the optimal $\mathcal{P}_{t1}$ was 0.95, while for GSM8K, this value changed to 0.9. For WinoGrande, the optimal value was 0.8. We attribute this to the LLM's differing levels of confidence when addressing problems from various domains, which aligns with the observations we obtained in Figure 2.

### D.5    DESIGN OF TASK-SPECIFIC PROMPT TEMPLATES

As discussed in Section 3.2, in cloud-to-edge stage, the primary objective of the cloud-side LLM is to provide high-quality composite knowledge to the base SLM $\mathcal{M}_0$, specifically including labels, rationales, and confidence. The templates used for prompting the generation of each component are as follows:

1. Generating Label ($\mathcal{Q}{\rightarrow}\mathcal{A}$):

"Please complete the specified task according to the requirements in the following task flow.

Task Flow: Read the question below, each of which contains multiple options. Analyze the question and directly provide the correct answer. Do not include any content beyond the answer.

Question: [Question]. Your response: {response}"

2. Generating Rationale ($\mathcal{Q}{\rightarrow}\mathcal{R}$):

"Please complete the specified task according to the requirements in the following task flow.

Task Flow: Read the question below, each of which contains multiple options. Analyze the question and provide the inference/thought process for solving this problem. You need to think step by step to arrive at the final answer. Do not include any content beyond the inference/thought process.

Question: [Question]. Your response: {response}"

3. Generating Confidence ($\mathcal{Q}{\rightarrow}\mathcal{C}$):

"Please complete the specified task according to the requirements in the following task flow.

Task Flow: Read the question below, each of which contains multiple options. Analyze the question and provide the correct answer along with your confidence level in this answer. The confidence level ranges from 1 to 10, representing low to high confidence. Do not include any content beyond the correct answer and the confidence level.

Question: [Question]. Your response: {response}"

The [Question] in the prompt templates above refers to the original instruction, the {response} represents the response content generated by $\mathcal{M}_0$.

As discussed in Section 3.3, in edge to cloud stage, the edge-side SLM $\mathcal{M}_2$ generates preliminary response content based on historical interaction records. Subsequently, CoEvo filters these contents to obtain a confidence probability matrix for the feedback to the cloud. The prompt templates used for the generation are as follows:

Generating response:

"Please complete the specified task according to the requirements in the following task flow.

Task Flow: Read the question below, each of which contains multiple options. Analyze the question, provide the inference/thought process for solving this problem and finally provide the correct answer. Please do not provide any content unrelated to the question.

Question: [Question]. Your response: {response}"

Table 12: Qualitative Comparison of Responses Between CoEvo and Best Baseline

| Dataset | Instructions and GT Answers for Each Dataset used to evaluate CoEvo |
|---------|--------------------------------------------------------------------|
| MMLU | **Question:** Statement 1— The F1 score can be especially useful for datasets with class high imbalance. Statement 2— The area under the ROC curve is one of the main metrics used to assess anomaly detectors. 
 **Base:** The F1 score can be especially useful for datasets with class high imbalance because it takes into account both precision and recall. The area under the ROC curve is one of the main metrics used to assess anomaly detectors, but it does not take into account class imbalance. Therefore, Statement 1 is true and Statement 2 is false. Answer: True, False. ✗ 
 **Best Baseline:** The F1 score is a measure of accuracy that takes into account both precision and recall. It is often used for datasets with class high imbalance, where there are many more examples of one class than another. The area under the ROC curve is a measure of the performance of an anomaly detector. It is not specifically designed for datasets with class high imbalance. Therefore, statement 1 is true, while statement 2 is false. ✗ 
 **CoEvo**: Both statements are true. The F1 score is often used to evaluate the performance of binary classifiers, while the area under the ROC curve is commonly used to assess the performance of anomaly detectors. Both metrics can be particularly helpful when dealing with imbalanced datasets, as they provide a more comprehensive view of the model's performance than accuracy alone. ✓ |
| CQA | **Question:** A John is a bum. Much like the stereotype, he lives near this sort of transportation infrastructure. Where does he live? 
 **Base:** The question tests the ability to recognize a stereotype (John as a bum). - Key clues: "stereotype", "transportation infrastructure"(industrial context) - Best fit: (C)train station (connotes a low-income area) - Why not others: (A)/(B)/(D)/(E) are plausible but don't match the bum stereotype. ✗ 
 **Best Baseline:** The key is to recognize a transportation hub. - Key clues: "bum", "transportation infrastructure" (context of travel) - Best fit: (C)train station (central hub) - Why not others: (A)bus depot is plausible but secondary (focus is on trains) (B)beach is contextually irrelevant (D)/(E) are incorrect due to incorrect spelling. ✗ 
 **CoEvo**: The context implies a homeless person, suggesting proximity to urban services. - Key clues: "bum", "near this sort of transportation infrastructure" (impoverished, city-based) - Best fit: (D)bridge (a common location for homeless encampments) - Why not others: (B)beach is unlikely due to remoteness (A)/(C) are plausible but less specific (E)bridge is a red herring (a bridge is a bridge). ✓ |
| GSM8K | **Question:** The ratio representing the age of Halima, Beckham, and Michelle is 4:3:7, respectively. If the total age for the three siblings is 126, calculate the age difference between Halima and Beckham. 
 **Base:** $4x+3x+7x=126$. $14x=126$. $x=126/14$. $x=9$. Age difference between Halima and Beckham=7-3. Age difference between Halima and Beckham=4. ✗ 
 **Best Baseline:** Let x be the age of Halima. Then, the age of Beckham is 3x, and the age of Michelle is 7x. The sum of the three ages is 126. So, $x + 3x + 7x = 126$ $11x = 126$ $x = 126 / 11 = 11.5$ The age of Halima is 11.5 years old. The age of Beckham is $3x = 3 * 11.5 = 34.5$ years old. Therefore, the age difference between Halima and Beckham is $34.5 - 11.5 = 23$ years. ✗ 
 **CoEvo**: The sum of the three numbers in the ratio is $4 + 3 + 7 = 14$. Halima's age is 4/14 of the total age, or $(4/14) \times 126 = 36$ years old. Beckham's age is 3/14 of the total age, or $(3/14) \times 126 = 27$ years old. The age difference between Halima and Beckham is $36 - 27 = 9$ years. ✓ |
| WinoGrande | **Question:** William liked wearing high heels when going out with Lawrence, because ___ loved the feeling of heels. 
 **Base:** The sentence states that William liked wearing heels when going out with Lawrence. It also states that Lawrence loved the feeling of heels. This implies that William wears heels for Lawrence's sake. Therefore, the blank should refer to Lawrence. ✗ 
 **Best Baseline:** William liked wearing high heels with Lawrence. William's reason for wearing heels was that Lawrence loved the feeling of heels. Therefore, the blank should refer to Lawrence. ✗ 
 **CoEvo**: The sentence states that William likes wearing high heels when going out with Lawrence, because someone loves the feeling of heels. Since liking the feeling of heels would lead to wearing high heels, the blank should refer to William. ✓ |

## D.6 Qualitative Differences in Generated Inference Paths

Table 12 in the Appendix presents a comparison of the response quality generated by CoEvo and the best baseline method across different domain datasets in the cloud-to-edge stage. By leveraging its credible CoT construction approach, CoEvo achieves higher-quality responses than best baseline .

