# OpenReview forum: "Breaking Static Paradigms: A Mutual Evolution Framework for Edge-Cloud Model Collaboration"
_ICLR.cc/2026/Conference — Submitted to ICLR 2026_

### Official Review · Reviewer_xwnm · 2025-10-21

**Soundness:** 1
**Presentation:** 2
**Contribution:** 1
**Rating:** 2
**Confidence:** 3

**Summary:**

This paper proposes CoEvo, a trainable framework for edge-cloud computing that finetunes both the cloud-based LLM and the edge-based LLM to improve performance. In particular, the LLM first distills knowledge in the SLM, weighting traces by their confidence of the LLM. Then, the SLM is first locally finetuned further and then used to update the LLM. For this purpose, the SLM generates probability matrices that are then sent and used by the LLM, only including matrices that have high probability generations. Results show that CoEvo outperforms baselines.

**Strengths:**

Unfortunately, I could not find any strengths of the work.

**Weaknesses:**

The paper suffers from several critical weaknesses and various weaknesses. I have split my concerns between these types and also included remarks, which are small points that have not influenced my judgment. I do note that due to many explanations missing (also a weakness), some of these points might come from misunderstanding on my part. The authors are free to refute them with a thorough explanation if they feel they are based on a misunderstanding.

**Critical Weaknesses**
- **The presented method seems far from applicable to any edge-based scenario**. The discrepancy between the proposed method and the proposed use-case are very high. Among others, here are a few indications that this method would never be practical:
   - If one can finetune an 8B model on edge-devices, one can also run a 70B model there. The compute required for finetuning are way higher than for inference, but the authors make no note of this, pretending it is easy for an edge-device to finetune an 8B model, which is most definitely not the case.
   - The evaluation does not actually do anything that is relevant to edge-based scenarios. The evaluation takes place on some standard benchmarks and an end-to-end evaluation is never performed. None of the benchmarks are somewhat applicable to any real-world scenario I can think of for edge-based devices (would be surprising if one user has GSM8k data, one has MMLU data, ...).
   - The framework is limited to final-answer or multiple-choice based benchmarks. This limits its application significantly, especially because the best use case for edge-based computing is personalization, which cannot be done with multiple-choice answers.
   - There is nothing in the paper that explains how to relegate complex queries from the SLM to the LLM. This is a key missing aspect for the application the authors are arguing about. Without it, we essentially just have two separate models that do their own thing.
- **The method significantly increases communication overhead and does not improve user privacy, despite claims.** The problem regards the size of the probability matrix: for a typical vocab size of 30,000 and relatively short reasoning response of 100 tokens, the authors would send 3 **million** floats to the cloud for each samples on which they want to train (approx 10MB). Furthermore, the data sample can be simply obtained from the probability matrix by argmaxxing each column, it does not give any extra privacy guarantees.
- **Performance gains are minimal.** Performance improvements are not noteworthy and less than 2% almost everywhere.

**Weaknesses**
- **The paper uses outdated models, techniques, and datasets.** None of the used datasets are really used in practice anymore. The models used are also outdated. Furthermore, techniques such as CoT are outdated, this technique has now been finetuned specifically into models. Requiring models to answer in a single token without CoT is outdated and should never be used anymore, except (maybe) for simple classification tasks (which for instance GSM8k is not).
- **Explanation about many critical aspects are missing.** There is an enormous amount of explanations missing that need to be filled out. For instance,
   - It is never explicitly mentioned what the training data is. I am currently assuming this is the training data associated with each benchmark? This would not be great, see later weakness.
   - It is not specified what training method is used for several components. For instance, the self-training of the SLM from $M_1$ to $M_2$ is not specified, other than that it is done. What is the data used here? On what traces is it finetuned? Does it do normal SFT? If so, how are the labels obtained (in most edge-based practical scenarios, labels are not available, since users only ask questions to LLMs that they don't know the answer to). Furthermore, it is unspecified what the loss function is that is used to train the LLM in the edge-to-cloud scenario: KL-divergence with respect to the probability matrix?
  - The setup for the paper is not properly identified: What are the assumption on the compute for the cloud and edge devices, what kinda tasks is the method applicable to? For instance, in section 3 the authors talk about labels without first having properly explained the method is only applicable to classification datasets.
  - Are the answers and reasoning in any way mixed in training the SLM in (3.2)? Since CoEvo only requiring a single inference step to classify things, what is the use of training on reasoning traces?
   - In 3.2, it is never mentioned how confidence is measured. Only in the appendix, it becomes clear from the prompt that this is self-reported confidence with some custom prompt. Furthermore, the prompt in this appendix asks to also provide the answer. How is this answer used? Does it need to match the answer provided in the single-step Q->A?
- **3.2 is not very novel, as it is a standard distillation technique.** The method distils the LLM into the SLM with some extra selection based on confidence. However, the plot in 2 (and the general literature) indicate that this is a rather poor selection method; confidence is only slightly higher for correct answer.
- **Training takes place on training sets (?) of benchmarks.** LLMs and SLMs should be used to handle a variety of questions. While an evaluation setup that contains a training set and test set with the exact same distribution is fine as a toy setup, it is not really applicable to any real use case.
- **Extra inferences required for the scheme significantly increase compute requirements.** Edge-devices are compute constraints. Yet, the method seems to not take this into account. Not only is finetuning directly performed on edge-nodes, majority voting is also performed on each sample, requiring many more inferences compared to standard operation procedures. No analysis of this extra compute requirement is mentioned or discussed.

**Questions:**

See above

---

> ### Author Response · Authors · 2025-11-25
>
> > **W1. Fine-tuning LLM on edge devices is currently infeasible.**
>
> **R1:**  Thank you for your valuable feedback.  We recognize that this stems from an insufficiently clear **definition and scope of the term "edge device"** in the manuscript. We sincerely apologize for this lack of clarity previously and provide a detailed elaboration and clarification.
>
> In fact, "edge side" is not limited solely to smartphones with relatively constrained computational resources. Our concept of the "edge" broadly includes **edge computing nodes with substantial computational capabilities**, such as high performance development kits(e.g., NVIDIA Jetson series, embedded devices with discrete GPUs), personal computers/workstations, onboard intelligent computing platforms, and small servers deployed at the edge. These devices are typically equipped with powerful CPUs, GPUs, or dedicated AI accelerators, possessing sufficient computational power to support lightweight fine-tuning of models with billions of parameters.
>
> Existing Parameter-Efficient Fine-Tuning (PEFT) techniques significantly reduce the computational cost of model adaptation.
>
> |        Method         |   Model   |    VRAM    |
> | :-------------------: | :-------: | :--------: |
> | Full Parameter Tuning | Llama3-8B |    68GB    |
> |         Lora          | Llama3-8B |   41.9GB   |
> |         QLora         | Llama3-8B | **23.2GB** |
>
> Compared to full fine-tuning, these methods can lower the VRAM requirement for a model like Llama3-8B to under 24GB. This makes it feasible to run on edge devices equipped with GPUs similar to an RTX 3090.
>
>
>
>
>
> > **W2.  Relies solely on standard academic benchmarks that do not represent realistic edge computing scenarios or user data distributions.**
>
> **R2:**  Thank you very much for raising this question. In the early stages of our research, it is necessary to follow established conventions by validating our approach on recognized standard benchmarks such as GSM8K and MMLU. These benchmarks provide an **academically recognized and widely used platform** for comparison. Furthermore, observing consistent performance improvements on standard benchmarks provides strong validation to our research.
>
> We will add a section in the Conclusion and Future Work part of the paper to explicitly state the proof of concept nature of the current work and emphasize that a key next step for subsequent research is dedicated to end-to-end evaluation in real-world scenarios.
>
>
>
> > **W3.  Fixed-answer benchmarks fundamentally limits its applicability to the core edge-computing use case of personalization.**
>
> **R3:**  Thank you for this important observation. Currently, suitable standardized real-world benchmarks based on edge-cloud architecture are lacking. Our work, as the first to explore mutual evolution in a cloud-edge collaboration setting, first establishes a algorithmic framework using controlled benchmarks to ensure comparability. We will certainly incorporate more realistic data in future work as the field matures.
>
>
>
> > **W4.  Lacks a critical mechanism for routing complex queries from SLM to LLM, undermining its core premise of collaborative inference.**
>
> **R4:**  Thank you for the question. Our work addresses a **different aspect** of edge-cloud collaboration. While many studies focus on real-time inference with cloud LLMs, our framework avoids such communication entirely during inference. Instead, the edge SLM operates independently after initial cloud distillation.
>
> Our key contribution is a **periodic co-evolution mechanism**: the edge model learns from local data and feeds knowledge back to the cloud, which then updates itself and distills an improved SLM for the next cycle. This enables continuous, bidirectional improvement without real-time cloud dependence.

---

> ### Author Response · Authors · 2025-11-25
>
> > **W5. How does probability matrix actually reduce communication volume compared to simpler alternatives?**
>
> **R5:**  Thank you for raising this concern. Before filtering the probability matrix obtained from inference, **CoEvo applies a compression process**. With the LLaMA3 series models, the tokenizer's vocabulary size is 128,000 tokens. However, through statistical analysis across the multiple datasets used in our experiments, we find that the probability distribution is highly concentrated in the **top-k** elements. We select the **top 10** elements, which significantly reduces the raw transmission cost while striving to preserve the uncertainty information inherent in the model's predictions.  We will add an explanation of the probability matrix preprocessing in Section 3.3 of the paper.
>
> Regarding the **comparison of transmission efficiency** between the probability matrix and raw data, the rationale within the edge-cloud architecture is as follows: traditional methods, when leveraging edge devices to acquire new data and subsequently update knowledge on both the edge and cloud sides, typically require **two** **data transmissions**. These are 1) transmitting the raw new data from the edge to the cloud, and 2) the cloud sending down updated data or model weights to update the edge model. CoEvo, by performing the update of the edge-side model locally, requires only **a single data transmission** from the edge to the cloud to complete an update cycle for the entire architecture, thereby reducing the number of transmissions by one. These explanations correspond to the analysis in the final paragraph of **Section 4.4** in our paper. We analyze the average instruction length of the different data types used in the experiments, calculate and test the size difference between “instruction + answer” and “instruction + compressed probability matrix”, ultimately yielding the results presented in **Table 5.**
>
> |         Method         |   CQA    |  GSM8K   | WinoGrande |
> | :--------------------: | :------: | :------: | :--------: |
> | w/o Local data storage |   2.0×   |   1.6×   |    1.8×    |
> |   Local data storage   |   1.3×   |   1.2×   |    1.2×    |
> |         CoEvo          | **1.0×** | **1.0×** |  **1.0×**  |
>
>  Therefore, CoEvo effectively reduces the communication overhead in the edge-cloud architecture.
>
>
>
> > **W6.  Probability matrix does not give any extra privacy guarantees.**
>
> **R6:**  Thank you for raising this important question. Our research focus is specifically on the **performance** and **efficiency** of the edge-cloud architecture, and privacy considerations are beyond its immediate focus.
>
> Privacy-enhancing techniques, such as differential privacy or homomorphic encryption, can subsequently be integrated as **enhancement modules** in future work, building upon the core architecture of our method.
>
>
>
> > **W7.  Performance gains are minimal.**
>
> **R7:**  Thank you for this critical comment. Contemporary LLMs like LLaMA3 and Qwen have achieved remarkably high performance on standard benchmarks. **Pushing the boundaries further is inherently difficult**, and even minor improvements are considered valuable in the field. Our goal is not merely to maximize performance at any cost in an idealized setting. Instead, we operate under the strict, **practical constraints** of a edge-cloud architecture, which involves a **LLM-to-SLM** knowledge transfer and aims to minimize communication overhead. Achieving any improvement under these challenging conditions, which include a significant model capacity gap and limited computational budget on the edge, could demonstrates the effectiveness of our co-evolutionary mechanism.
> While the gains are in the 1-2% range, they are positive and **consistent** across all datasets and model pairs we tested. This consistency is a key strength. It indicates that our method provides a reliable and robust boost, rather than producing erratic results that are highly sensitive to the dataset or model architecture.
>
> Note: We conduct a thorough re-examination of the root causes and perform additional experimental designs to address the suboptimal performance of our algorithm on the GSM8K dataset during the edge-to-cloud stage. The specific work undertaken and the latest experimental results are elaborated as follows.
>
> | Method | Dataset | Origin | Improved  |
> | ------ | ------- | ------ | --------- |
> | SPIN   | GSM8K   | 62.08  |  62.08   |
> | CoEvo  | GSM8K   | 60.77  | **62.44** |

---

> ### Author Response · Authors · 2025-11-25
>
> > **W8. Outdated models, techniques, and datasets.**
>
> **R8:**
>
> Thank you for raising this point.
>
> Research on **edge-cloud collaborative architectures** is indeed a current **hotspot**, attracting significant attention for its potential to balance capability and efficiency. However, as this direction still **lacks** established baselines, dedicated benchmarks, and specialized models, we adopted classic relevant baseline methods and datasets in our experimental design, along with the Llama 3 model, to ensure comparability and reproducibility.
>
> Considering the rapid iteration speed in the LLM field, we supplement our study with the latest model pair from the **Qwen series** (Qwen3-1.4B and Qwen3-17B) and conduct a series of experiments to validate our method.
>
> | Method(cloud-to-edge stage) | MMLU      |
> | --------------------------- | --------- |
> | Qwen3 1.7B                  | 56.97     |
> | Distill step by step        | 62.90     |
> | SPIN                        | 63.12     |
> | **CoEvo(ours)**             | **64.72** |
>
>
>
> | Method(edge-to-cloud stage) | CQA |
> | --------------------------- | ------------- |
> | Qwen3 14B                   | 81.80         |
> | **CoEvo(ours)**             | **83.04**     |
>
> Our new experimental results consistently demonstrate that our framework **remains effective** on the datasets evaluated using the Qwen model pair.
>
>
>
> > **W9.  Technical implementation is critically under-specified, including the SLM fine-tuning method, data source, label availability and the specific loss function used for LLM updates in the E2C stage.**
>
> **R9:**  Thank you for these insightful questions. Here are our explanations.
>
> **SLM fine-tuning method:**  The local update from M1 to M2 is indeed performed using Supervised Fine-Tuning (**SFT**).
>
> **Data source:** The training data is sourced from a split of the standard benchmark datasets (e.g., a portion of the GSM8K or WinoGrande training sets) used in our experiments. This controlled setup allows for a clear and comparable evaluation of our core algorithm.
>
> **Label availability:**  Our framework is built upon a key difference from traditional static edge-cloud architectures. Unlike traditional models where the edge device's role is largely passive (mainly for inference), our design explicitly assumes that the edge device is an active agent capable of continuously acquiring and learning from new local data. This perspective is inspired by the **client dynamics** in Federated Learning (FL), but with a different focus. While FL primarily aims to aggregate knowledge from distributed data without centralizing it, our co-evolutionary framework focuses on creating a **synergistic learning loop** between edge and the cloud, where each side not only contributes to but also benefits from the other's learning process.
>
> **Loss function:**  The Cloud LLM is fine-tuned using the knowledge (e.g., the compressed probability matrices) uploaded from the edge devices. The loss function used to incorporate this knowledge is indeed the Kullback-Leibler (**KL**) Divergence.

---

> ### Author Response · Authors · 2025-11-25
>
> > **W10.  Lacks critical setup specifications, including the assumed computational capabilities of cloud/edge devices and the specific class of tasks.**
>
> **R10:**  Thank you for your question. Here are our detailed explanations.
>
> **Computational capabilities:**  In fact, "edge side" is not limited solely to smartphones with relatively constrained computational resources. Our concept of the "edge" broadly includes **edge computing nodes with substantial computational capabilities**, such as high performance development kits(e.g., NVIDIA Jetson series, embedded devices with discrete GPUs), personal computers/workstations, onboard intelligent computing platforms, and small servers deployed at the edge. These devices are typically equipped with powerful CPUs, GPUs, or dedicated AI accelerators, possessing sufficient computational power to support lightweight fine-tuning of models with billions of parameters.
>
> Existing Parameter-Efficient Fine-Tuning (PEFT) techniques significantly reduce the computational cost of model adaptation.
>
> |        Method         |   Model   |    VRAM    |
> | :-------------------: | :-------: | :--------: |
> | Full Parameter Tuning | Llama3-8B |    68GB    |
> |         Lora          | Llama3-8B |   41.9GB   |
> |         QLora         | Llama3-8B | **23.2GB** |
>
> Compared to full fine-tuning, these methods can lower the VRAM requirement for a model like Llama3-8B to under 24GB. This makes it feasible to run on edge devices equipped with GPUs similar to an RTX 3090. **In our work, the term "edge devices" refers to those meeting or exceeding this specification.**
>
> **Specific class of tasks:**  In the experimental section, we indeed utilize datasets with ground-truth labels to validate our method on standard benchmarks for comparative clarity. However, the core mechanism of our approach does not rely on ground-truth labels. The supervisory signals used within the framework are derived from 1) **high quality answers and reasoning chains generated by the cloud LLM**, and 2) the **high confidence outputs of the edge-side model itself**. Consequently, the method can be naturally extended to open-domain tasks lacking ground-truth labels, such as personalized dialogue or creative writing. In these scenarios, the "labels" are the high quality content generated and selected by the LLM or SLM itself.
>
>
>
>
>
> > **W11.  Are the answers and reasoning in any way mixed in training the SLM in (3.2)? Since CoEvo only requiring a single inference step to classify things, what is the use of training on reasoning traces?**
>
> **R11:**  Thank you for this excellent question.
>
> The final answer and its rationale are not fused into a single output during training. Instead, they are generated through two separate, sequential reasoning steps, each driven by a distinct prompt.
>
> The use of reasoning traces (rationales) during training is a deliberate and crucial aspect of our method. While the final output is a single token or short answer, training the model on the complete reasoning chain provides a significantly **richer learning signal**. The rationale generated by the powerful Cloud LLM encapsulates its **thinking process** (logical steps, relevant knowledge) and problem solving strategy required to arrive at the answer. Distilling this rich contextual information is far more effective for teaching the small SLM than merely training it to mimic the final answer label. By training on the rationale, we teach the SLM **how to derive the answer, not just to memorize it**.
>
>
>
> > **W12.  How confidence is quantified and whether the answer generated alongside it in the prompt is used?**
>
> **R12:**  Thank you for pointing out these problems. The confidence score is obtained through a custom prompt eliciting self-reported confidence from the model.
>
> Meanwhile, the answer generated alongside the confidence score must be **consistent** with the answer produced in the single-step Q→A inference. This consistency is crucial for **maintaining the integrity of the training data**. It ensures that the confidence score is being calibrated against the same output that the small model (SLM) is ultimately trained to produce. Any significant discrepancy would introduce noise and undermine the reliability of the confidence-based filtering and weighting mechanisms in our training objective.

---

> ### Author Response · Authors · 2025-11-25
>
> > **W13.  Confidence is potentially poor, and it's only slightly higher for correct answer.**
>
> **R13:**  Thank you for this comment. In the design of our work, we did thoroughly consider exploring more methods to measure rationale quality. Ultimately, we chose to focus on **optimizing** and **utilizing** the confidence scores generated by the LLM itself. Here is the detailed explanation.
>
> The advantage of the confidence metric lies in its **ease of acquisition**. It also concisely represents the LLM's self certainty, making it particularly **suited** for edge-cloud collaboration scenarios.
>
> We design a specific, decoupled confidence prompting strategy for complex inference tasks like GSM8K to improve the accuracy of their assessment. This represents an exploration into how to use the LLM to produce more reliable quality signals.
>
> Furthermore, we **explore various designs for how the confidence score is applied** within the training objective function (as mentioned in Appendix D.2 of the paper), ensuring that this single score can effectively guide the training process.
>
> |       Methods        |   MMLU    |    CQA    |   GSM8K   | WinoGrande |
> | :------------------: | :-------: | :-------: | :-------: | :--------: |
> |    Meta Llama3 8B    |   57.26   |   59.38   |   52.15   |   62.46    |
> | Distill step by step |   64.38   |   65.44   |   59.31   |   68.82    |
> |         M1−A         | **65.91** | **66.26** | **60.77** |   68.71    |
> |         M1−B         |   64.28   |   64.79   |   59.20   |   67.88    |
> |         M1−C         |   63.60   |   63.25   |   57.11   | **70.40**  |
>
> M1-A, M1-B, and M1-C represent different training objectives we explored. The results show that for different tasks, only minor adjustments to the use of the confidence metric are needed to achieve results **surpassing** the baseline, demonstrating the effectiveness of our approach.
>
>
>
> > **W14.  Training on training sets (?) of benchmarks is not really applicable to any real use case.**
>
> **R14:**  Thank you for raising this important point. In our experiments, the **knowledge flow involves cross-dataset transfer**. Specifically, the cloud-to-edge distillation is performed using one dataset, while local updates on the edge device are conducted on a different dataset that simulates the device's local data environment.
>
> This setup is intentionally designed to be more challenging and realistic than a standard i.i.d. split. It tests the framework's ability to not just memorize a dataset, but to leverage **generalized knowledge** distilled from the cloud LLM and then adapt it to a **new task** on the edge. Results under this cross-dataset setting provide strong evidence for the method's applicability to scenarios where the edge data distribution differs from the cloud's initial training data.

---

> ### Author Response · Authors · 2025-11-25
>
> > **W15.  Multi-stage pipeline and sophisticated components introduce significant implementation complexity, but  no analysis is mentioned or discussed.**
>
> **R15:**  Thank you for your attention. We conduct tests on the computational overhead required for several steps executed by the edge-side model during the edge-to-cloud stage and compare it with approaches such as 'transmitting data only ' and '"update + transmitting data"'.
>
> |           Method           | Local optimization | Multiple-sample voting | Probability matrices filtering | Total |
> | :------------------------: | :----------------: | :--------------------: | :----------------------------: | :---: |
> |   transmitting data only   |         0          |           0            |               0                |   0   |
> | update + transmitting data |        3.9h        |         5mins          |               0                | 3.98h |
> |           CoEvo            |        3.9h        |         26mins         |               1s               | 4.32h |
>
> "Local optimization " refers to the process from M1 to M2, while " multiple-sample voting " denotes the process where the edge-side model of CoEvo generates the original knowledge uploaded to the cloud. After balancing latency and performance considerations, CoEvo uses a sampling count of **5**. This means that compared to the original inference, it introduces approximately **4 times the additional** computational overhead for inference.
>
> Before filtering the probability matrix obtained from inference, CoEvo actually **compresses** it. LLaMA3 models' vocabulary size of the tokenizer is 128,000. However, based on statistical analysis across multiple datasets used in our experiments, we find that the probability distribution is concentrated in the top-k elements. We select the top 10 elements, which significantly reduces the original transmission cost while striving to preserve the uncertainty information of the model's predictions.
>
> We will add a new subsection in Section 4 of the paper to elaborate on the aforementioned analysis of computational overhead.
>
> In the ablation experiments shown in Figure 3, we explore the **impact** of different sampling counts on inference accuracy.
>
> | Ablation ID | Multiple-sample voting | Probability matrices filtering |    CQA    |   GSM8K   | WinoGrande |
> | :---------: | :--------------------: | :----------------------------: | :-------: | :-------: | :--------: |
> |      1      |           No           |               No               |   78.64   |   79.19   |   81.13    |
> |      2      |          Yes           |               No               |   78.71   |   80.76   |   83.72    |
> |      3      |           No           |              Yes               |   78.84   |   80.44   |   83.50    |
> |  **CoEvo**  |          Yes           |              Yes               | **79.26** | **81.18** | **83.79**  |
>
> The "**update + transmitting data**" method mentioned earlier corresponds to the sample with Ablation ID 1. As can be seen, it not only fails to significantly reduce the computational overhead on the device side but also considerably **degrades** the performance of cloud-side optimization.
>
> Regarding the "**transmitting data only**" method, while this method avoids edge computational costs, it comes at the cost of increased communication complexity, as the updated cloud LLM must provide additional feedback to the edge SLM for their co-evolution.
>
> |         Method         |   CQA    |  GSM8K   | WinoGrande |
> | :--------------------: | :------: | :------: | :--------: |
> | w/o Local data storage |   2.0×   |   1.6×   |    1.8×    |
> |   Local data storage   |   1.3×   |   1.2×   |    1.2×    |
> |         CoEvo          | **1.0×** | **1.0×** |  **1.0×**  |
>
> We have analyzed the associated overhead in Section 4.4, and Table 5 presents the results.  This strategy requires two data transmissions between the edge and cloud to achieve a collaborative update. Even with local storage of new data, it still necessitates the additional transmission of the cloud-based LLM’s response data to update the edge-side SLM.

---

> > ### Comment · Reviewer_xwnm · 2025-11-26
> > **Reply to Rebuttal**
> >
> > I thank the authors for their rebuttal, but find it overall unconvincing and thereby retain my score. In particular:
> >
> > 1. Any point that is not clarified in the paper, I consider unresolved. Since (almost) no updates were made to the paper, most points remain unresolved.
> >
> > 2. A tiny performance improvement is not appreciated by the field. This makes the method at the very most incremental, but potentially just noise. I don't think any person would consider applying this method in a practical setting, given that the toy benchmarks that are used in a very controlled setting only improve performance by 1%, which is quite close to nothing given the differences in effort (setting this all up compared to doing nothing).
> >
> > 3. Several aspects of the algorithm rely on the benchmarks being final-answer ones. Thereby, it is impossible to apply it to relevant benchmarks (R3).
> >
> > 4. I really do not see any application for this. Not only are the evaluated settings just not realistic, but if none of the data is private, it is much simpler to just train the (small) models on the server, rather than using isolated client-nodes with very limited compute to do the training.
> >
> > 5. The authors are performing QLora on the client nodes, this is never mentioned in the text.
> >
> > 6. None of my remarks are actually addressed in the rebuttal to a degree I would retract the remark (except the communication overhead and the privacy remark).

---

> > > ### Author Response · Authors · 2025-11-28
> > > **Thank you for your comment**
> > >
> > > > **Any point that is not clarified in the paper, I consider unresolved. Since (almost) no updates were made to the paper, most points remain unresolved.**
> > >
> > > Thank you for your response! We apologize for any insufficient revisions. We have now uploaded a new revised version, with **the newly modified content highlighted in red**.
> > >
> > > The main changes are as follows:
> > >
> > > 1. We revised the description of the critical challenge and motivation, removing potentially confusing content and incorporating the issue of data transfer overhead into our motivation.
> > > 2. In Section 3.3, we elaborated on the compression strategy for the probability matrix.
> > > 3. In Section 4.1, we provided more detailed experimental setups, including computational resource usage and training strategies.
> > > 4. We substantially revised Section 4.4 to focus more on the analysis of generalizability and comparisons derived from computational overhead analysis, while removing the chart analysis related to communication overhead.
> > > 5. Added experiments on heterogeneous model pairs in Appendix D.1.
> > >
> > > Due to the overall complexity of the experiment, we don't explore the use of more efficient fine-tuning techniques in the current work, and we will continue to expand their application in future studies.
> > >
> > > >**A tiny performance improvement is not appreciated by the field.**
> > >
> > > Thank you for your concern. Our experimental results on each dataset are obtained from multiple **repeated** runs (no fewer than four times). Furthermore, while ensuring identical experimental conditions, we synchronously adjusted certain training parameters for both the baseline methods and our approach across multiple trials and still observed a consistent performance gap. This provides strong evidence that the results are not due to random noise but represent a stable and reproducible improvement.
> > >
> > > Although the absolute improvement of our method over the baseline at each stage is modest, our bidirectional knowledge transfer strategy in the edge-cloud scenario enables the stable derivation of an optimized, high-performance small model from a large cloud-based model, while also allowing the personalized small model on the edge side to successfully enhance the large cloud-based model. Therefore, the improvement we achieve actually stems from cumulative gains across both stages, representing a consistent and stable performance enhancement of the entire edge-cloud architecture.
> > >
> > > > **Several aspects of the algorithm rely on the benchmarks being final-answer ones. Thereby, it is impossible to apply it to relevant benchmarks (R3).**
> > >
> > > Thank you for your response. The reason for using these general benchmarks in our current experiments is that they provide a **reliable means** of validating model inference performance. These well-established benchmarks have been extensively validated and are frequently used as baselines for evaluating new models.
> > >
> > > During the evaluation of CoEvo, we also analyzed **perplexity**, a metric reflecting the certainty of model outputs, and presented the results in Appendix Table 10.
> > >
> > > | Methods(cloud-to-edge stage) | MMLU     | CQA      | GSM8K    | WinoGrande |
> > > | ---------------------------- | -------- | -------- | -------- | ---------- |
> > > | Llama3 8B                    | 4.49     | 10.26    | 6.77     | 3.47       |
> > > | CoT                          | 4.08     | 9.44     | 6.56     | 3.29       |
> > > | self-consistency CoT         | 4.08     | 9.44     | 6.56     | 3.29       |
> > > | ToT                          | 3.40     | 8.08     | 5.94     | 3.01       |
> > > | DPO                          | 3.37     | 7.43     | 5.67     | 2.87       |
> > > | Distill step by step         | 3.25     | 6.82     | 5.98     | 3.06       |
> > > | SPIN                         | 3.28     | 7.52     | 5.91     | 3.03       |
> > > | CoEvo                        | **2.90** | **6.51** | **5.60** | **2.59**   |
> > >
> > > | Method(edge-to-cloud stage) | CQA      | GSM8K    | WinoGrande |
> > > | --------------------------- | -------- | -------- | ---------- |
> > > | Llama3 70B                  | 7.91     | 8.43     | 2.27       |
> > > | CoEvo                       | **7.65** | **7.99** | **2.22**   |
> > >
> > > Our method improves inference performance while reducing perplexity, indicating that the model exhibits greater certainty in its inference outputs. This further validates the effectiveness of our approach.

---

> > > > ### Author Response · Authors · 2025-11-28
> > > >
> > > > >**I really do not see any application for this.**
> > > >
> > > > Thank you for this comment. In real-world edge-cloud scenarios, the number of edge devices is often **massive**. When the cloud side handles data requests or parameter distribution for a large number of devices, it inevitably faces significant communication load. This load can easily lead to **delays** or even failures on the cloud side, negatively impacting user experience. The relatively limited resources of individual edge devices **collectively** represent a substantial computational resource pool. Our approach **shifts** a large portion of the computation to local devices, which can significantly alleviate the computational and communication pressure on the cloud (since we cannot assume that cloud resources are infinitely abundant), thus enhances the operational efficiency of the edge-cloud ecosystem and improves the overall user experience.
> > > >
> > > > > **The authors are performing QLora on the client nodes, this is never mentioned in the text.**
> > > >
> > > > Thank you for your concern. We acknowledge that QLoRA is not used in the current experiments; instead, LoRA was employed. Our reference to QLoRA in the response was intended to illustrate that edge devices can achieve considerable LLM fine-tuning with techniques like PEFT, thus extending the discussion. However, in our experiments involving edge-side models, computational demands were strictly controlled. We apologize for any confusion caused by our previous reply.
> > > >
> > > > Due to the overall complexity of the experiment, we don't explore the use of more efficient fine-tuning techniques in the current work, and we will continue to expand their application in future studies.
> > > >
> > > > > **None of my remarks are actually addressed in the rebuttal to a degree I would retract the remark (except the communication overhead and the privacy remark).**
> > > >
> > > > Thank you for raising this point. As mentioned earlier, we have submitted the updated manuscript and addressed several of your concerns. We apologize for the previous insufficient explanations and the need for manuscript revisions, and we hope the updated content could resolves your questions.

---

### Official Review · Reviewer_ZPEU · 2025-10-25

**Soundness:** 3
**Presentation:** 3
**Contribution:** 3
**Rating:** 6
**Confidence:** 3

**Summary:**

This paper addresses the limitation of static knowledge in current edge-cloud collaborative language model architectures, where powerful LLMs reside in the cloud and smaller SLMs operate on edge devices. Existing methods for updating these systems often involve retraining the cloud LLM with raw data collected from the edge, incurring high communication costs, latency, privacy risks, and underutilizing edge compute capabilities. To overcome this, the authors propose CoEvo, a framework enabling mutual evolution where both the cloud LLM and edge SLMs continuously update with new knowledge. Experiments using Llama3 8B (edge) and 70B (cloud) on datasets like MMLU, CQA, GSM8K, and WinoGrande show that CoEvo improves the performance of the edge SLM compared to various baselines and also enhances the cloud LLM's performance compared to its base version, often achieving gains comparable to non-private SFT.

**Strengths:**

The core contribution is the introduction of a bidirectional, continuous learning framework for edge-cloud collaboration, moving beyond static deployments or simple cloud-centric updates. This mutual evolution concept is novel and addresses a significant limitation of prior work. The framework directly tackles major issues in edge-cloud systems: incorporating new knowledge dynamically, reducing communication overhead, enhancing privacy, utilizing edge compute resources for local updates, and mitigating catastrophic forgetting in the cloud LLM. Experiments demonstrate clear performance improvements on both the edge SLM (outperforming strong CoT and fine-tuning baselines) and the cloud LLM (achieving gains comparable to non-private SFT, while better preserving general capabilities). The ablation studies effectively validate the contribution of the key components (confidence weighting, sampling, filtering).

**Weaknesses:**

CoEvo involves multiple stages (cloud-to-edge, edge-to-cloud) and several sophisticated sub-components (confidence scoring, CoT generation, majority voting, probability matrix extraction, dual-threshold filtering). This introduces significant implementation complexity compared to simpler update strategies. The cloud-to-edge distillation relies heavily on the LLM's self-reported confidence being a reliable indicator of correctness/quality. While Figure 2 provides some justification, confidence calibration in LLMs is known to be imperfect and can vary across models and domains. Poor calibration could negatively impact the SLM distillation.

**Questions:**

How sensitive is the cloud-to-edge distillation performance to the calibration quality of the LLM's confidence scores? Have you experimented with different methods for obtaining or scaling these confidence scores?

---

> ### Author Response · Authors · 2025-11-25
>
> > **W1.  Multi-stage pipeline and sophisticated components introduce significant implementation complexity compared to simpler strategies.**
>
> **R1：**  Thank you for your attention. We conduct tests on the computational overhead required for several steps executed by the edge-side model during the edge-to-cloud stage and compare it with approaches such as 'transmitting data only ' and '"update + transmitting data"'.
>
> |           Method           | Local optimization | Multiple-sample voting | Probability matrices filtering | Total |
> | :------------------------: | :----------------: | :--------------------: | :----------------------------: | :---: |
> |   transmitting data only   |         0          |           0            |               0                |   0   |
> | update + transmitting data |        3.9h        |         5mins          |               0                | 3.98h |
> |           CoEvo            |        3.9h        |         26mins         |               1s               | 4.32h |
>
> "Local optimization " refers to the process from M1 to M2, while " multiple-sample voting " denotes the process where the edge-side model of CoEvo generates the original knowledge uploaded to the cloud. After balancing latency and performance considerations, CoEvo uses a sampling count of **5**. This means that compared to the original inference, it introduces approximately **4 times the additional** computational overhead for inference.
>
> Before filtering the probability matrix obtained from inference, CoEvo actually **compresses** it. LLaMA3 models' vocabulary size of the tokenizer is 128,000. However, based on statistical analysis across multiple datasets used in our experiments, we find that the probability distribution is concentrated in the top-k elements. We select the top 10 elements, which significantly reduces the original transmission cost while striving to preserve the uncertainty information of the model's predictions.
>
> We will add a new subsection in Section 4 of the paper to elaborate on the aforementioned analysis of computational overhead.
>
> In the ablation experiments shown in Figure 3, we explore the **impact** of different sampling counts on inference accuracy.
>
> | Ablation ID | Multiple-sample voting | Probability matrices filtering |    CQA    |   GSM8K   | WinoGrande |
> | :---------: | :--------------------: | :----------------------------: | :-------: | :-------: | :--------: |
> |      1      |           No           |               No               |   78.64   |   79.19   |   81.13    |
> |      2      |          Yes           |               No               |   78.71   |   80.76   |   83.72    |
> |      3      |           No           |              Yes               |   78.84   |   80.44   |   83.50    |
> |  **CoEvo**  |          Yes           |              Yes               | **79.26** | **81.18** | **83.79**  |
>
> The "**update + transmitting data**" method mentioned earlier corresponds to the sample with Ablation ID 1. As can be seen, it not only fails to significantly reduce the computational overhead on the device side but also considerably **degrades** the performance of cloud-side optimization.
>
> Regarding the "**transmitting data only**" method, while this method avoids edge computational costs, it comes at the cost of increased communication complexity, as the updated cloud LLM must provide additional feedback to the edge SLM for their co-evolution.
>
> |         Method         |   CQA    |  GSM8K   | WinoGrande |
> | :--------------------: | :------: | :------: | :--------: |
> | w/o Local data storage |   2.0×   |   1.6×   |    1.8×    |
> |   Local data storage   |   1.3×   |   1.2×   |    1.2×    |
> |         CoEvo          | **1.0×** | **1.0×** |  **1.0×**  |
>
> We have analyzed the associated overhead in Section 4.4, and Table 5 presents the results.  This strategy requires two data transmissions between the edge and cloud to achieve a collaborative update. Even with local storage of new data, it still necessitates the additional transmission of the cloud-based LLM’s response data to update the edge-side SLM.

---

> ### Author Response · Authors · 2025-11-25
>
> > **W2&W3&Q4.**  **Vulnerable to the miscalibration of LLM confidence scores, raising concerns about its robustness and the need to explore alternative confidence-setting methods.**
>
> **R2:**  Thank you for your concern. While confidence metric has been questioned in some previous works, our method effectively leverages it to appropriately balance the training objectives, ultimately achieving superior performance on the test datasets. As shown in Table 1, CoEvo outperforms the variant without the confidence metric (Distill step by step).
>
> |        Method        |   MMLU    |    CQA    |   GSM8K   | WinoGrande |
> | :------------------: | :-------: | :-------: | :-------: | :--------: |
> | Distill step by step |   64.38   |   65.44   |   59.31   |   68.20    |
> |      **CoEvo**       | **65.91** | **66.26** | **60.77** | **70.40**  |
>
> This demonstrates that the appropriate application of the confidence metric can yield beneficial effects in edge-cloud architecture optimization.
>
> Furthermore, we **explore various designs** for how the confidence score is applied within the training objective function (as mentioned in Appendix D.2 of the paper), ensuring that this single score can effectively guide the training process.
>
> |       Methods        |   MMLU    |    CQA    |   GSM8K   | WinoGrande |
> | :------------------: | :-------: | :-------: | :-------: | :--------: |
> |    Meta Llama3 8B    |   57.26   |   59.38   |   52.15   |   62.46    |
> | Distill step by step |   64.38   |   65.44   |   59.31   |   68.82    |
> |         M1−A         | **65.91** | **66.26** | **60.77** |   68.71    |
> |         M1−B         |   64.28   |   64.79   |   59.20   |   67.88    |
> |         M1−C         |   63.60   |   63.25   |   57.11   | **70.40**  |
>
> M1-A, M1-B, and M1-C represent different training objectives we explored. The results show that for different tasks, only minor adjustments to the use of the confidence metric are needed to achieve results **surpassing** the baseline, demonstrating the effectiveness of our approach.

---

### Official Review · Reviewer_zRaQ · 2025-10-28

**Soundness:** 1
**Presentation:** 2
**Contribution:** 2
**Rating:** 2
**Confidence:** 3

**Summary:**

The paper proposes CoEvo, a bidirectional edge-cloud learning framework. Different from the traditional point of view, where the edge uploads data to the cloud, CoEvo considers a self-evolutional strategy on the edge. In the cloud-to-edge phase, an LLM distills to an on-device SLM using a confidence-weighted CoT loss. In the edge-to-cloud phase, the SLM uploads filtered probability matrices using dual thresholds on top-k mass and max probability. Experiments on four benchmarks show modest improvements over inference baselines and small gains over strong finetuning baselines.

**Strengths:**

- This work presents an intuitive idea that is easy to implement.
- Results show that CoEvo outperforms most inference baselines and is competitive with SFT.

**Weaknesses:**

- I don't see this approach to be much practical.

First of all, we shouldn't expect edge devices to do fine-tuning by themselves. Edge devices are resource-constrained devices, and available compute power should be prioritized for inference. Even for finetuning, we can expect those updates to be done in a non-real-time manner (such as during maintenance or during less-service hours). I don't see a need to address the immediate latency issue. Second, the privacy of the user data is also not guaranteed. The probability matrices essentially reveal the user data, and I don't see how the data can be protected. Third, I'd say that matrices are much more communication-intensive than raw data.

- The idea of selective distillation is not novel.

In previous works [1], [2], this idea has been proposed.

[1] Chen et al. Towards Robust and Efficient Cloud-Edge Elastic Model Adaptation via Selective Entropy Distillation, ICLR 2024.

[2] Shao et al. Selective Knowledge Sharing for Privacy-Preserving Federated Distillation without A Good Teacher, Nature Communications.

**Questions:**

1. What is the exact payload size per edge-to-cloud update? How efficient is it to transmit a matrix compared to raw data?
2. Can you run membership inference or input reconstruction attacks against uploaded matrices to check if the claim over user privacy is held?
3. Please address the weakness. In particular, why is there a need for an immediate edge-side update, and why can't we let more powerful cloud services update edge models, while only transmitting user data from edge to cloud and model weights from cloud to edge?

---

> ### Author Response · Authors · 2025-11-25
>
> > **W1. Fine-tuning LLM on edge devices is currently infeasible (or Fine-tune in a non-real-time manner).**
>
> **R1:**  Thank you for your comment. Upon analysis, we recognize that this comment originate in an insufficiently clear **definition and scope of the term "edge device"** in our manuscript. We sincerely apologize for this lack of clarity and would like to provide a detailed explanation.
>
> In our research, the constraint on the computational resources of "edge devices" is relative. While edge devices are unsuitable for directly deploying massive language models, they are capable of running and optimizing smaller models. Our concept of the "edge" broadly includes**edge computing nodes with considerable computational capabilities**, such as high-performance development boards (e.g., NVIDIA Jetson series, embedded devices with discrete GPUs), personal computers/workstations, onboard intelligent computing platforms, and small servers deployed at the edge. These devices are typically equipped with powerful CPUs, GPUs, or dedicated AI accelerators, possessing sufficient computing power to support lightweight fine-tuning of models with billions of parameters.
>
> Existing Parameter-Efficient Fine-Tuning (PEFT) techniques significantly reduce the computational cost of model adaptation.
>
> |        Method         |   Model   |    VRAM    |
> | :-------------------: | :-------: | :--------: |
> | Full Parameter Tuning | Llama3-8B |    68GB    |
> |         Lora          | Llama3-8B |   41.9GB   |
> |         QLora         | Llama3-8B | **23.2GB** |
>
> Compared to full fine-tuning, these methods can lower the VRAM requirement for a model like Llama3-8B to under 24GB. This makes it feasible to run on edge devices equipped with GPUs similar to an RTX 3090.
>
> Regarding **Fine-tune in a non-real-time manner**, in edge computing scenarios, model updates should indeed be non-real-time tasks performed using idle resources. We apologize for the misunderstanding caused by our failure to explicitly emphasize this objective in the paper.
>
> The goal is to utilize the computational resources of the devices during their idle, low-load periods, thereby completely avoiding any impact on the user experience and improving computational resource utilization. We have revised the manuscript accordingly and will add a new subsection in Section 4 of the paper to elaborate on the aforementioned analysis of computational overhead, emphasizing that our method is designed precisely for **periodic, offline fine-tuning scenarios**.
>
>
>
> > **W2&Q2. Claim of user privacy is questionable.**
>
> **R2:**  Thank you for raising this important question. Our research focus is specifically on the performance and efficiency of the edge-cloud architecture, and privacy considerations are beyond its immediate focus.
>
> Privacy-enhancing techniques, such as differential privacy or homomorphic encryption, can subsequently be integrated as **enhancement modules** in future work, building upon the core architecture of our method.

---

> ### Author Response · Authors · 2025-11-25
>
> > **W3&Q1.  Communication benefit is unclear, as probability matrices appear more data-intensive than raw inputs, requires a comparison of payload sizes.**
>
> **R3:**  Thank you for raising this concern. Before filtering the probability matrix obtained from inference, **CoEvo applies a compression process**. For instance, with the LLaMA3 series models, the tokenizer's vocabulary size is 128,000 tokens. However, through statistical analysis across the multiple datasets used in our experiments, we find that the probability distribution is highly concentrated in the **top-k** elements. We select the **top 10** elements, which significantly reduces the raw transmission cost while striving to preserve the uncertainty information inherent in the model's predictions.  We will add an explanation of the probability matrix preprocessing in Section 3.3 of the paper.
>
> Regarding the **comparison of transmission efficiency** between the probability matrix and raw data, the rationale within the edge-cloud architecture is as follows: traditional methods, when leveraging edge devices to acquire new data and subsequently update knowledge on both the edge and cloud sides, typically require **two** **data transmissions**. These are 1) transmitting the raw new data from the edge to the cloud, and 2) the cloud sending down updated data or model weights to update the edge model. CoEvo, by performing the update of the edge-side model locally, requires only **a single data transmission** from the edge to the cloud to complete an update cycle for the entire architecture, thereby reducing the number of transmissions by one. These explanations correspond to the analysis in the final paragraph of **Section 4.4** in our paper. We analyze the average instruction length of the different data types used in the experiments, calculate and test the size difference between “instruction + answer” and “instruction + compressed probability matrix”, ultimately yielding the results presented in **Table 5.**
>
> |         Method         |   CQA    |  GSM8K   | WinoGrande |
> | :--------------------: | :------: | :------: | :--------: |
> | w/o Local data storage |   2.0×   |   1.6×   |    1.8×    |
> |   Local data storage   |   1.3×   |   1.2×   |    1.2×    |
> |         CoEvo          | **1.0×** | **1.0×** |  **1.0×**  |
>
>  Therefore, CoEvo effectively reduces the communication overhead in the edge-cloud architecture.
>
>
>
> > **W4.  The idea of selective distillation is not novel.**
>
> **R4:**  Thank you for this important question. We have carefully studied this paper and note that there are several differences between our work and the papers mentioned.
>
> Regarding **work [1]**, we also note this paper during our initial design phase. Our objective is to address the novel challenges of edge-cloud collaborative learning in the era of LLM. Specifically, we focus on how an edge-side SLM can perform continuous learning **using local data**, efficiently feed this knowledge back to the cloud, and how the device and cloud can achieve **efficient collaborative updates** to reduce communication overhead. In contrast, work [1] addresses the **dynamic adaptation of traditional visual models on edge devices**, primarily exploring how edge-cloud collaboration can enhance the robustness of edge models against distribution shifts.
>
> Furthermore, the method in [1] updates the **parameters of the normalization layers** in the edge model. While this is common and effective for traditional models, it is insufficient in the context of LLM. In CoEvo, edge side only needs to upload **lightweight information** to the cloud.
>
> Finally, the operational paradigm in [1] remains **cloud-centric**, where the cloud model leads and updates the edge model. Our work, conversely, establishes a **bidirectional flow of knowledge.** Not only does the cloud model provide high quality knowledge, but the edge model can also provide feedback to the cloud after assimilating new knowledge.
>
> Regarding **work [2]**, the motivation is to improve the **traditional federated distillation mechanism** by enhancing distillation efficiency, particularly under **non-IID data distributions**, through knowledge selection between clients and the server. The context of our work is the inherent **communication overhead** and the performance **gap** between the models in edge-cloud collaborative architecture. To tackle this, we modify the traditional cloud-based update mechanism by considering the edge device's capability to acquire new knowledge and performing local updates on the edge model. This **shifts** part of the workload traditionally located in the cloud to the edge, significantly reducing the communication overhead caused by frequent edge-cloud data transfers. Through mechanisms like credible chain of thought (CoT) knowledge distillation, multi-sample voting, and probability matrix filtering, the specialized SLM on the edge can provide high quality, domain-specific knowledge to feedback and enhance cloud LLM.

---

> ### Author Response · Authors · 2025-11-25
>
> > **Q3.  Design of edge updates requires justification against a simpler alternative: sending raw data to the cloud and receiving updated model weights.**
>
> **R5:**  Thank you for your attention. The primary reason for not adopting the mentioned approach is that it would introduce significant additional **communication overhead** and underutilize the computational capabilities of the **edge devices**.
>
> **Communication overhead:**  Traditional methods, when leveraging edge devices to acquire new data and subsequently update knowledge on both the edge and cloud sides, typically require **two** **data transmissions**. These are 1) transmitting the raw new data from the edge to the cloud, and 2) the cloud sending down updated data or model weights to update the edge model. CoEvo, by performing the update of the edge-side model locally, requires only **a single data transmission** from the edge to the cloud to complete an update cycle for the entire architecture, thereby reducing the number of transmissions by one. These explanations correspond to the analysis in the final paragraph of **Section 4.4** in our paper. We analyze the average instruction length of the different data types used in the experiments, calculate and test the size difference between “instruction + answer” and “instruction + compressed probability matrix”, ultimately yielding the results presented in **Table 5.**
>
> |         Method         |   CQA    |  GSM8K   | WinoGrande |
> | :--------------------: | :------: | :------: | :--------: |
> | w/o Local data storage |   2.0×   |   1.6×   |    1.8×    |
> |   Local data storage   |   1.3×   |   1.2×   |    1.2×    |
> |         CoEvo          | **1.0×** | **1.0×** |  **1.0×**  |
>
>  Therefore, CoEvo effectively reduces the communication overhead in the edge-cloud architecture.
>
> **Edge devices:**  Traditional cloud-based update approach fails to account for the substantial idle computational resources present in extensive edge devices within real-world scenarios. Due to intermittent user activity, these edge devices often remain low load or idle for considerable periods. Although cloud resources are abundant, the conventional method, which requires the cloud to update and distribute parameters to all edge devices, can still incur significant latency when handling requests from a massive number of devices. In contrast, CoEvo offloads most update computations to these edge devices, effectively leveraging their idle capacity while easing the computational burden on the cloud.

---

> > ### Comment · Reviewer_zRaQ · 2025-11-25
> > **Thank you for your rebuttal**
> >
> > I acknowledge the recipient of the rebuttal.
> >
> > After reading the rebuttal, I still have some concerns. I'll follow the author's list and address each point.
> >
> > > W1:
> >
> > conditionally resolved. Please update the paper and let me know once you've done so.
> >
> > > W2&Q2:
> >
> > This response contradicts some of the central claims in the paper. In my review, I treated those claims of privacy protection as part of the contribution. Please remove those claims of enhanced privacy. For example, the claim of "secure" in line 92, the claims of data privacy around line 431, etc.
> >
> > > W3&Q1:
> >
> > I'm not fully following the logic here. Even though the matrix is a top-10 compression, we still need 10 tokens for 1 token of raw data. That might not be a big deal depending on the data size, but it is not more efficient than sending the raw data. Table 5 does not really compare transmission between the matrix and the raw data. If I understand correctly, it compares a cloud-based update vs an edge-based self-update. So, in short, I can see how an edge self-update is better than a cloud update due to efficiency, but the rationale for matrix transmission over raw data transmission seems to lie simply in a little bit of better generalization preservation (Table 4?).
> >
> > > W4:
> >
> > I still hold my previous judgment that the novelty is not obvious. It still seems to be a confidence-based selective distillation. But don't worry, I do not penalize the work much for lack of novelty. I gave a low score because the work was confusing.
> >
> > > Q3:
> >
> > Resolved.
> >
> > > Overall:
> >
> > The rebuttal has addressed some of my confusions. I maintain the score for now since the authors did not update the PDF, and I think the paper needs substantial revision. If the authors can rewrite the paper to reflect their promises and remove the overclaims, I'm happy to increase my score.

---

> > > ### Author Response · Authors · 2025-11-26
> > >
> > > Thank you for carefully reviewing our response and providing further feedback! As for the revised manuscript, we **highlight them in blue** font in the main text for your convenience.
> > >
> > >
> > >
> > > > **W1: Conditionally resolved. Please update the paper and let me know once you've done so.**
> > >
> > > **Q1:**  Thank you for the reminder! We have revised and updated the manuscript, supplementing it with an introduction to edge devices.
> > >
> > >
> > >
> > > > **W2&Q2: Please remove those claims of enhanced privacy.**
> > >
> > > **Q2:**  Thank you for your response. We have rechecked the manuscript and removed all statements regarding privacy.
> > >
> > >
> > >
> > > > **W3&Q1:  Need to clarify the comparison between the matrix and the original data. The advantage of the matrix lies in its better generalization performance.**
> > >
> > > **Q3:**  Thank you for your concern. We apologize for the lack of clarity in our previous response. The analysis of the results in Table 5 is indeed a comparison of transmission overhead between CoEvo and **traditional cloud-based update methods**. As you correctly pointed out, the probability matrix output by the model is indeed larger in size compared to hard-label answers. In the context of edge-based self-update, the reason for adopting the probability matrix is that it provides better **generalization** performance, thereby ensuring that the cloud-side LLM does not lose its general capabilities, which is crucial for the overall evolution of the edge-cloud architecture.
> > >
> > >
> > >
> > > > **W4: Novelty is not obvious.**
> > >
> > > **Q4:**  Thank you so much for your response! We aim to provide an explanation that focuses specifically on the innovative aspects of our work. In previous responses, we placed emphasis on **comparative analysis** with existing works you mentioned.
> > >
> > > For CoEvo, its innovation stems from the edge-cloud collaborative architecture it is built upon, which represents a current **research frontier** and has become a hot topic in major forums and conferences. At the same time, due to the forward-looking nature of this direction, there are currently **few mature works** based on the edge-cloud architecture, and our study is the **first** to explore bidirectional knowledge transfer and collaborative evolution mechanisms within such a framework.
> > >
> > > Furthermore, by offloading a significant portion of computational tasks to edge devices with untapped potential, our method substantially alleviates the **critical communication latency** issue in edge-cloud scenarios. Through mechanisms such as bidirectional knowledge distillation, inference voting, and matrix filtering, we address the **performance gap** between large cloud-based models and smaller edge-based models. Specifically, the domain-specific personalized model on the edge can provide high-quality knowledge to the large cloud-based model.

---

> > > > ### Comment · Reviewer_zRaQ · 2025-11-26
> > > > **Please be serious about reviewer's comments and your own work**
> > > >
> > > > > We have revised and updated the manuscript.
> > > >
> > > > When I say the paper needs substantial revision, I mean it. Substantial revision does not mean just adding a paragraph and deleting a few words here and there. Have you even reread your own work? Do you think it looks like a coherent piece? As of now, the methodological content might be mostly correct, but the consistency of the story is nowhere to be found. For example, line 66, "directly uploading raw data to the cloud for updating LLMs incurs substantial communication overhead." Do you think this is a correct statement? Newly added line 73, "NVIDIA Jetson series ... possessing sufficient computing power to support lightweight fine-tuning of models with billions of parameters." This is technically true, but you need to consider that your experiments are done on A100. In the rebuttal, you say k=10. Why don't you update the paper to say k=10? Reviewer xwnm also raised a good point. What is your PEFT strategy? If you use QLoRA, why don't you say so in the paper? If you didn't, why do you use it to support your paper in the rebuttal?
> > > >
> > > > Section 4.4 remains confusing. Now you've given up the claim on data efficiency per transmission and state the benefit as generalization improvement, why do you still keep Table 5 in Section 4.4? Table 5 should be your motivation to do edge-side self-update, and should be put somewhere much earlier. Also, what is "X-based" in line 457?
> > > >
> > > > There are probably many more writing issues that I have not found out about. **The authors really need to be serious about their own work.**

---

> > > > > ### Author Response · Authors · 2025-11-28
> > > > >
> > > > > Thank you for your response! Due to time constraints, in our previous reply and during the revision of the paper, we primarily focused on the **specific** modification points you directly mentioned. We apologize for any insufficient revisions. We have now uploaded a new revised version, with **the newly modified content highlighted in red**.
> > > > >
> > > > > The main changes are as follows:
> > > > >
> > > > > 1. We revised the description of the critical challenge and motivation, removing potentially confusing content and incorporating the issue of data transfer overhead into our motivation.
> > > > > 2. In Section 3.3, we elaborated on the compression strategy for the probability matrix.
> > > > > 3. In Section 4.1, we provided more detailed experimental setups, including computational resource usage and training strategies.
> > > > > 4. We substantially revised Section 4.4 to focus more on the analysis of generalizability and comparisons derived from computational overhead analysis, while removing the chart analysis related to communication overhead.
> > > > > 5. Added experiments on heterogeneous model pairs in Appendix D.1.
> > > > >
> > > > > Regarding **QLoRA**: We acknowledge that QLoRA is not used in the current experiments; instead, LoRA was employed. Our reference to QLoRA in the response was intended to illustrate that edge devices can achieve considerable LLM fine-tuning with techniques like PEFT, thus extending the discussion. However, in our experiments involving edge-side models, computational demands were strictly controlled. We apologize for any confusion caused by our previous reply.
> > > > >
> > > > > Due to the overall complexity of the experiment, we don't explore the use of more efficient fine-tuning techniques in the current work, and we will continue to expand their application in future studies.

---

### Official Review · Reviewer_7Ggt · 2025-10-31

**Soundness:** 3
**Presentation:** 3
**Contribution:** 2
**Rating:** 4
**Confidence:** 4

**Summary:**

This work proposes CoEvo, a mutual evolution framework for edge-cloud model collaboration that enables continuous learning of new knowledge. It uses credible Chain-of-Thought distillation to transfer knowledge from cloud LLMs to edge SLMs, and credible probability matrices to feedback domain-specific knowledge from edge to cloud, achieving performance improvements on both sides while reducing communication overhead.

**Strengths:**

+ CoEvo introduces a mutual evolution framework for edge-cloud collaboration, breaking static paradigms by enabling both cloud LLMs and edge SLMs to continuously learn and improve from new domain-specific knowledge dynamically.
+ The framework incorporates confidence scores to weight and filter knowledge transfer in both directions, ensuring only high-quality information is distilled. This credibility mechanism mitigates the impact of erroneous outputs and improves training effectiveness.
+ CoEvo reduces data transmission overhead by performing local updates on edge devices and uploading only filtered probability matrices instead of raw data, achieving comparable performance to centralized training approaches.

**Weaknesses:**

-  Technical contribution is limited considering the existing work of mutual knowledge distillation between cloud and edge, such as COLLA proposed by Lu et al. in SEC 2019. The core techniques—confidence-weighted knowledge distillation and probability matrix filtering—are incremental extensions of existing mutual distillation methods. The framework primarily combines established techniques rather than introducing fundamentally novel mechanisms for edge-cloud collaboration.

- Fine-tuning billion-parameter models (Llama3-8B) on memory-constrained edge devices like smartphones is currently infeasible. The paper lacks discussion of computational requirements, memory constraints, and practical deployment challenges on real edge hardware.

- Experiments only evaluate one model pair (Llama3-8B/70B) across four datasets. The generalizability to other model architectures, size ratios, and diverse application domains remains unvalidated, limiting confidence in the framework's broad applicability.

- The paper lacks thorough analysis of critical design choices, such as convergence behavior across multiple iterations, robustness to noisy edge data, and failure modes. The filtering thresholds appear dataset-specific without principled selection guidelines.

**Questions:**

Please see Weaknesses.

---

> ### Author Response · Authors · 2025-11-25
>
> > **W1.  Core technical contribution appears incremental, as it primarily combines established mutual distillation techniques (Like COLLA).**
>
> **R1:** Thank you very much for your problem. We have carefully studied this paper and found there are **fundamental differences** between our work and COLLA in terms of the motivation addressed, the technical pathway. These differences, we argue, constitute the key novelty of our paper.
>
> Regarding the **motivation**, COLLA focuses on addressing **data heterogeneity in user behavior prediction** within a edge-cloud architecture. Its goal is to use collaborative learning to train personalized, lightweight models for each device. In contrast, CoEvo addresses the **massive communication overhead** generated by large language model (LLM) services in a edge-cloud setup. Therefore, our focus is on how to enable edge devices to efficiently and cost-effectively leverage the capabilities of cloud-based LLMs, rather than relying on the static update mechanism from the cloud.
>
> As for the **technical pathway**, in COLLA, whether the cloud distributes an initial model or devices upload updated models, the transmitted content consists of the **entire model's parameters**. This approach was viable in the era of small models but is inappropriate for large models. Conversely, in CoEvo, the edge devices do not need to upload any model parameters whatsoever, thereby eliminating the problem of transmitting massive parameter sets. The edge side only needs to **upload lightweight information to the cloud**, such as user queries and the inference results (represented with uncertainty) from the local model on the device. The cloud then uses these soft labels from the edge to fine-tune or enhance the knowledge of the LLM. This process constitutes a continuous optimization of the model's knowledge, differing from COLLA's approach of aggregating small models.
>
> In summary, while the **contribution** of COLLA lies in proposing a collaborative learning method for data-heterogeneous user behavior prediction tasks, our work introduces a mutual evolution mechanism based on bidirectional knowledge transfer within a edge-cloud architecture. This approach breaks conventional cloud-based static updating paradigm, establishing a more dynamic and interactive co-evolution process between the edge and cloud.
>
>
>
>
>
> > **W2.  Fine-tuning LLM on edge devices like smartphones is currently infeasible.**
>
> **R2:**  Thank you for your comment. We recognize that this stems from an insufficiently clear **definition and scope of the term "edge device"** in the manuscript. We sincerely apologize for this lack of clarity previously and provide a detailed elaboration and clarification.
>
> In fact, "edge side" is not limited solely to smartphones with relatively constrained computational resources. Our concept of the "edge" broadly includes **edge computing nodes with substantial computational capabilities**, such as high performance development kits(e.g., NVIDIA Jetson series, embedded devices with discrete GPUs), personal computers/workstations, onboard intelligent computing platforms, and small servers deployed at the edge. These devices are typically equipped with powerful CPUs, GPUs, or dedicated AI accelerators, possessing sufficient computational power to support lightweight fine-tuning of models with billions of parameters.
>
> Existing Parameter-Efficient Fine-Tuning (PEFT) techniques significantly reduce the computational cost of model adaptation.
>
> |        Method         |   Model   |    VRAM    |
> | :-------------------: | :-------: | :--------: |
> | Full Parameter Tuning | Llama3-8B |    68GB    |
> |         Lora          | Llama3-8B |   41.9GB   |
> |         QLora         | Llama3-8B | **23.2GB** |
>
> Compared to full fine-tuning, these methods can lower the VRAM requirement for a model like Llama3-8B to under 24GB. This makes it feasible to run on edge devices equipped with GPUs similar to an RTX 3090.

---

> ### Author Response · Authors · 2025-11-25
>
> > **W3.  Lack of experimental validation on other heterogeneous model pairs.**
>
> **R3:** Sincere thanks for raising the concern. Our focus on the Llama3-8B/70B pair is primarily because its prominence as a community standard offers **significant influence and broad applicability**, allowing for in-depth analysis on a mature benchmark.
>
> Furthermore, we now conduct extensive supplementary experiments involving a different model family. Specifically, we evaluate our framework on the Qwen3-1.7B and Qwen3-14B models.
>
> | Method(cloud-to-edge stage) |   MMLU    |
> | :-------------------------: | :-------: |
> |         Qwen3 1.7B          |   56.97   |
> |    Distill step by step     |   62.90   |
> |            SPIN             |   63.12   |
> |       **CoEvo(ours)**       | **64.72** |
>
>
>
> | Method(edge-to-cloud stage) | CQA |
> | :-------------------------: | :-----------: |
> |          Qwen3 14B          |     81.80     |
> |       **CoEvo(ours)**       |   **83.04**   |
>
> Our new experimental results consistently demonstrate that our framework **remains effective** on the datasets evaluated using the Qwen model pair. This positive outcome strongly supports the universality of our method across different model architectures and scale ratios.
>
> We will incorporate these new experiments, along with the corresponding analysis and discussion, into Section 4.3 of the revised manuscript.
>
>
>
> > **W4.  Lacks analysis of critical design choices (e.g., convergence, robustness), and the use of dataset-specific thresholds without principled guidelines.**
>
> **R4:** Thank you for your feedback. We carefully address each of your points by supplementing the paper with additional statistics and in-depth analysis.
>
> **Convergence:**  We record the loss values during training.
>
> | Loss  | epoch1 | epoch2 | epoch3 | epoch4 | epoch5 |
> | ----- | ------ | ------ | ------ | ------ | ------ |
> | train | 1.19   | 0.92   | 0.54   | 0.42   | 0.39   |
> | eval  | 1.17   | 0.77   | 0.55   | 0.52   | 0.51   |
>
> Results show that CoEvo exhibits a relatively smooth and stable convergence trend on the experimental datasets. The training and eval loss continuously decreases, while the performance shows steady improvement and gradually stabilizes. This demonstrates the robustness of our training procedure.
>
> **Robustness:**  In fact, the datasets used in our experiments **inherently contain noise**, as they comprise labels, rationales, and confidence scores generated by the teacher model. The fact that CoEvo still achieves stable performance improvements on these datasets indicates a good level of robustness to such data noise.
>
> **Failure mode:**   The explicit analysis of failure modes is not included in the current study, as our primary focus is on establishing and validating the mutual evolution mechanism based on edge-cloud architecture. A systematic failure mode analysis for this architecture presents **significant challenges**. The primary difficulties lie in the precise definition and triggering of specific failure cases in a controlled manner. We acknowledge that investigating failure cases is a **valuable direction**, and we will incorporate a dedicated analysis into our future research.
>
> **Thresholds:**  We provide an experimental analysis of different thresholds in Figure 5 in Appendix. Results indicate that while the absolute optimal threshold varies slightly across datasets, the performance remains stably superior to the baseline as long as the threshold is **within a reasonable range** (specifically, the **0.8-0.95** interval explored). This suggests that CoEvo's effectiveness is not critically dependent on a precise threshold value and demonstrates a degree of robustness to the threshold selection.

---

### Official Review · Reviewer_Rsbi · 2025-11-01

**Soundness:** 3
**Presentation:** 4
**Contribution:** 4
**Rating:** 8
**Confidence:** 4

**Summary:**

This paper addresses the static nature of current edge-cloud model collaboration frameworks, where Large Language Models (LLMs) on the cloud and Small Language Models (SLMs) on the edge cannot be dynamically updated with new knowledge. The authors highlight that existing update methods, which involve uploading new edge-side data for retraining, suffer from high communication overhead, user latency, and data privacy concerns.

To solve this, the paper proposes CoEvo (Collaboration-Evolution), a novel mutual evolution framework. The framework consists of two main stages: Cloud-to-Edge: Credible CoT Knowledge Distillation & Edge-to-Cloud: Credible Probability Matrix Knowledge Distillation.

**Strengths:**

The paper introduces a novel two-stage "mutual evolution" framework that is well-designed and addresses a clear gap in existing edge-cloud architectures.
A major strength is the edge-to-cloud update mechanism. By optimizing the SLM locally and then uploading only filtered, "credible probability matrices" instead of raw data, the framework provides a practical way to learn from new domain-specific data while preserving user privacy.
The paper provides strong empirical support for its claims. It not only shows performance gains against strong baselines (Table 1) but also includes crucial ablation studies (Table 3) and a direct comparison against SFT (Table 4) to demonstrate its ability to mitigate catastrophic forgetting.

**Weaknesses:**

In the cloud-to-edge stage (Table 1), CoEvo underperforms the SPIN baseline on the GSM8K dataset. The paper hypothesizes this is due to the confidence metric struggling with tasks where multiple valid reasoning paths (CoTs) exist. This suggests the "credible CoT" mechanism may be less effective for domains with high solution entropy or creative reasoning tasks.
The edge-to-cloud stage requires the "resource-limited edge device" to perform several computationally non-trivial steps: local optimization (training M1 into M2), "multiple-sample voting" (Figure 3 suggests 10-20 samples), and then calculating and filtering probability matrices. The paper focuses on communication overhead savings but does not sufficiently analyze the computational overhead this process adds to the edge device.

**Questions:**

Could the authors elaborate on the computational overhead (e.g., latency, memory usage) of the edge-to-cloud pre-processing stage? Specifically, how does the cost of local optimization (M1 -> M2), multi-step sampling, and matrix filtering compare to a simple inference on the edge device? This seems like a significant workload for a "resource-limited" device.
Given the C2E stage's weaker performance on GSM8K and the hypothesis that this is due to multiple valid CoTs, does this limit CoEvo's applicability in more open-ended or creative tasks? Have the authors considered alternative methods for assessing rationale quality beyond a single confidence score for these domains?

---

> ### Author Response · Authors · 2025-11-25
>
> > **W1&Q2.  CoEvo's underperformance on GSM8K suggests its "credible CoT" mechanism may limit its applicability to open-ended or creative reasoning tasks.**
>
> **R1:** Thanks for your constructive comment. The suboptimal performance of CoEvo on GSM8K stems from the **particularity** of this dataset, which leads to a higher sensitivity in the confidence metric of our method. Here is our explanation and supplementary analysis regarding this issue.
>
> Specifically, the confidence metric appears to be more **sensitive** to question sequences that require multi-step calculations and involve complex symbolic reasoning. We further analyze GSM8K datasets , where generating step-by-step calculations and reasoning processes is often necessary before arriving at the final answer.
>
> Meanwhile, we revisit the detailed experimental design and identify the **root cause**: unlike general datasets that only require direct answer generation, GSM8K necessitates that the model outputs a **complete reasoning chain** (rationale), which includes mathematical calculation steps. Initially, we apply a uniform method for obtaining confidence scores across all datasets, i.e., prompting the model to generate both "answer + rationale" and "answer + confidence" simultaneously. This method relies on the strong correlation between the answer and the rationale to indirectly assess the credibility of the rationale.
>
> However, we observe that on GSM8K, when the model is asked to assess its confidence while simultaneously generating a lengthy rationale, **it tends to produce generally low confidence scores**, which corresponds to the statistical results in Figure 2. We hypothesize that this occurs because generating a long rationale inherently involves greater uncertainty and complexity, leading the model to provide a conservative estimate of its overall credibility. This systematic low confidence bias subsequently **affect** the confidence based optimization algorithm in CoEvo, ultimately leading to suboptimal performance.
>
> To address this issue, we modify the confidence score acquisition mechanism for the GSM8K dataset. The new method separates the generation of the rationale from the assessment of confidence. First, the model generates the complete "rationale + final answer". Subsequently, we take this complete output from the first step and incorporate it into a new prompt, directly asking the model to score the credibility of this content. Experiments show that this method effectively improves the accuracy and reasonableness of the confidence scores.
>
> | method | dataset | origin | improved  |
> | :----: | :-----: | :----: | :-------: |
> |  SPIN  |  GSM8K  | 62.08  |   62.08   |
> | CoEvo  |  GSM8K  | 60.77  | **62.44** |
>
> After implementing the new confidence acquisition method, **CoEvo's performance on the GSM8K dataset has improved significantly**. CoEvo now surpasses the SPIN baseline on this dataset, successfully resolving the previous performance bottleneck. Although the current margin of improvement over the baseline on this dataset is not as large as its performance on other domain-specific datasets, this enhancement further validates the effectiveness of the "credible CoT" mechanism.

---

> ### Author Response · Authors · 2025-11-25
>
> > **W2&Q1.  Computational overhead of the edge-to-cloud stage seems significant for a resource-limited device and requires further analysis.**
>
> **R2:** Thank you for your attention to this issue. In our context, edge devices include those with **certain computational capabilities**, such as Jetson developer kits and personal computers/small servers equipped with GPUs. Given our definition of capable edge devices, the introduced computational overhead is **acceptable** and does not pose a problem. Here we present a comparative analysis of CoEvo and several simplified methods in terms of their computational/communication overhead and performance.
>
> We conduct tests on the computational overhead required for several steps executed by the edge-side model during the edge-to-cloud stage and compare it with approaches such as 'transmitting data only ' and '"update + transmitting data"'.
>
> |           Method           | Local optimization | Multiple-sample voting | Probability matrices filtering | Total |
> | :------------------------: | :----------------: | :--------------------: | :----------------------------: | :---: |
> |   transmitting data only   |         0          |           0            |               0                |   0   |
> | update + transmitting data |        3.9h        |         5mins          |               0                | 3.98h |
> |           CoEvo            |        3.9h        |         26mins         |               1s               | 4.32h |
>
> "Local optimization " refers to the process from M1 to M2, while " multiple-sample voting " denotes the process where the edge-side model of CoEvo generates the original knowledge uploaded to the cloud. After balancing latency and performance considerations, CoEvo uses a sampling count of **5**. This means that compared to the original inference, it introduces approximately **4 times the additional** computational overhead for inference.
>
> Before filtering the probability matrix obtained from inference, CoEvo actually **compresses** it. LLaMA3 models' vocabulary size of the tokenizer is 128,000. However, based on statistical analysis across multiple datasets used in our experiments, we find that the probability distribution is concentrated in the top-k elements. We select the top 10 elements, which significantly reduces the original transmission cost while striving to preserve the uncertainty information of the model's predictions.
>
> We will add a new subsection in Section 4 of the paper to elaborate on the aforementioned analysis of computational overhead.
>
> In the ablation experiments shown in Figure 3, we explore the **impact** of different sampling counts on inference accuracy.
>
> | Ablation ID | Multiple-sample voting | Probability matrices filtering |    CQA    |   GSM8K   | WinoGrande |
> | :---------: | :--------------------: | :----------------------------: | :-------: | :-------: | :--------: |
> |      1      |           No           |               No               |   78.64   |   79.19   |   81.13    |
> |      2      |          Yes           |               No               |   78.71   |   80.76   |   83.72    |
> |      3      |           No           |              Yes               |   78.84   |   80.44   |   83.50    |
> |  **CoEvo**  |          Yes           |              Yes               | **79.26** | **81.18** | **83.79**  |
>
> The "**update + transmitting data**" method mentioned earlier corresponds to the sample with Ablation ID 1. As can be seen, it not only fails to significantly reduce the computational overhead on the device side but also considerably **degrades** the performance of cloud-side optimization.
>
> Regarding the "**transmitting data only**" method, while this method avoids edge computational costs, it comes at the cost of increased communication complexity, as the updated cloud LLM must provide additional feedback to the edge SLM for their co-evolution.
>
> |         Method         |   CQA    |  GSM8K   | WinoGrande |
> | :--------------------: | :------: | :------: | :--------: |
> | w/o Local data storage |   2.0×   |   1.6×   |    1.8×    |
> |   Local data storage   |   1.3×   |   1.2×   |    1.2×    |
> |         CoEvo          | **1.0×** | **1.0×** |  **1.0×**  |
>
> We have analyzed the associated overhead in Section 4.4, and Table 5 presents the results.  This strategy requires **two** data transmissions between the edge and cloud to achieve a collaborative update. Even with local storage of new data, it still necessitates the additional transmission of the cloud-based LLM’s response data to update the edge-side SLM. In CoEvo, edge only needs to transmit a portion of the new data and its own responses to the cloud.

---

> ### Author Response · Authors · 2025-11-25
>
> > **Q3.**  **Have the authors considered alternative methods for assessing rationale quality beyond a single confidence score for these domains?**
>
> **R3:** Thank you for your question. In the design of our work, we did thoroughly consider exploring more methods to measure rationale quality. Ultimately, we chose to focus on **optimizing** and **utilizing** the confidence scores generated by the LLM itself. Here is the detailed explanation.
>
> The advantage of the confidence metric lies in its **ease of acquisition**. It also concisely represents the LLM's self certainty, making it particularly **suited** for edge-cloud collaboration scenarios.
>
> As detailed in our response to the previous comment, we design a specific, decoupled confidence prompting strategy for complex inference tasks like GSM8K to improve the accuracy of their assessment. This represents an exploration into how to use the LLM to produce more reliable quality signals.
>
> Furthermore, we **explore various designs for how the confidence score is applied** within the training objective function (as mentioned in Appendix D.2 of the paper), ensuring that this single score can effectively guide the training process.
>
> |       Methods        |   MMLU    |    CQA    |   GSM8K   | WinoGrande |
> | :------------------: | :-------: | :-------: | :-------: | :--------: |
> |    Meta Llama3 8B    |   57.26   |   59.38   |   52.15   |   62.46    |
> | Distill step by step |   64.38   |   65.44   |   59.31   |   68.82    |
> |         M1−A         | **65.91** | **66.26** | **60.77** |   68.71    |
> |         M1−B         |   64.28   |   64.79   |   59.20   |   67.88    |
> |         M1−C         |   63.60   |   63.25   |   57.11   | **70.40**  |
>
> M1-A, M1-B, and M1-C represent different training objectives we explored. The results show that for different tasks, only minor adjustments to the use of the confidence metric are needed to achieve results **surpassing** the baseline, demonstrating the effectiveness of our approach.

---

### Meta-Review · Area_Chair_SFPq · 2026-01-07

**Summary:**

CoEvo proposes a mutual-evolution edge–cloud collaboration framework where a cloud LLM distills “credible” CoT knowledge to edge SLMs, and edge SLMs locally adapt on new data and then update the cloud using a filtered/top-k “probability matrix” signal (instead of uploading full raw data), aiming to reduce communication while enabling continual improvement on both sides.

As the claimed contributions regarding improved privacy and communication efficiency were criticized during the rebuttal and the scope of the paper had to narrow in response, another round of review after substantial changes would be good before acceptance at a major machine learning venue.

**Reviewer Concerns:**

1. Cloud→edge stage underperforms baseline on GSM8K; “credible CoT/confidence” may fail on multi-step / high-entropy reasoning tasks [Rsbi] Addressed? Mostly. Authors diagnose a prompting artifact (confidence asked while generating long rationales yields systematically low confidence), switch to a two-step rationale-then-score prompt, and report GSM8K improving from 60.77 → 62.44, surpassing SPIN (62.08).

2. Edge→cloud stage seems computationally heavy for “resource-limited” devices (local optimization + multi-sample voting + filtering); missing latency/memory analysis [Rsbi, ZPEU, 7Ggt, xwnm] Addressed? Partially. Authors provide a breakdown (e.g., ~3.9h local optimization + 26min voting + ~1s filtering; total ~4.32h) and clarify they target capable edge nodes (Jetson-class dev kits / PCs / small GPU servers) and that updates can be offline/idle-time, but this still leaves a gap vs the paper’s original “resource-limited edge” framing and does not provide on-device measurements for truly constrained devices (phones/sensors).

3. Novelty is incremental / largely combines established mutual distillation / selective distillation; related work overlap (e.g., COLLA; selective distillation papers) [7Ggt, zRaQ] Addressed? Partially. Authors argue CoEvo differs by targeting LLM-era communication/parameter-size constraints and by not transmitting model parameters (only lightweight signals), and emphasize bidirectional knowledge flow. This helps positioning, but reviewers who view it as “confidence-based selective distillation” may remain unconvinced.

4. Practicality concerns: edge devices shouldn’t fine-tune; if you can fine-tune 8B on edge, why not run bigger models; unclear assumptions about edge/cloud compute [7Ggt, zRaQ, xwnm] Addressed? Partially. Authors narrow the definition of “edge” to GPU-capable nodes (e.g., RTX 3090 class), cite PEFT VRAM estimates (LoRA/QLoRA) and stress offline updates. This resolves some misunderstanding but also weakens the original edge narrative and still doesn’t show feasibility on the kinds of edge devices many reviewers had in mind.

5. Communication benefit unclear: probability matrices may be larger than raw data; payload size not clearly reported; Table 5 logic confusing [zRaQ, xwnm] Addressed? Partially, with a major reframing. Authors initially argue top-k (k=10) compression and fewer transmissions; reviewer zRaQ pushes back that matrix > raw answer. Authors eventually concede: matrix isn’t smaller than hard labels, and the motivation becomes better generalization / preserving cloud general capabilities, while “fewer transmissions” comes from edge self-update vs cloud update. This is a meaningful clarification, but it signals earlier claims were overstated and the story/table placement was confusing.

6. Privacy claims are unsupported; probability matrices can leak inputs; request membership inference / reconstruction testing [zRaQ, xwnm] Addressed? Mixed (mostly by removing the claim). Authors explicitly state privacy is out of scope and agree to remove privacy/secure claims from the paper after reviewer pressure. This resolves “overclaiming,” but does not provide the requested privacy evidence and may reduce the paper’s claimed contribution scope.

7. Confidence-score calibration reliability is questionable; sensitivity to how confidence is obtained/scaled [ZPEU, Rsbi, xwnm] Addressed? Partially. Authors show CoEvo outperforms a “distill step-by-step” variant and provide several alternative objective variants (M1-A/B/C) plus special handling for GSM8K (decoupled prompt). However, they do not provide a direct calibration/sensitivity study (e.g., temperature scaling, external calibration, confidence noise injection), so robustness to miscalibration remains somewhat open.

8. Limited experiments: only one model pair and few datasets; unclear generality [7Ggt] Addressed? Mostly. Authors add experiments on another family/pair (Qwen3 1.7B/14B) and show improvements, which directly targets this concern (though breadth is still limited).

9. Under-specified methodology: training data, local SLM update details, loss for edge→cloud update, PEFT strategy, etc. [xwnm] Addressed? Partially-to-mostly. Authors clarify: local update is SFT, data comes from benchmark splits, cloud update uses KL divergence to the uploaded distributions, and explain confidence prompting/consistency checks. But reviewer feedback indicates the paper text lagged behind rebuttal claims (e.g., k=10, resources, training strategy), and writing coherence remained a serious issue.

10. Benchmarks / setting not realistic for edge personalization; “routing” mechanism between SLM/LLM missing; unclear end-to-end edge-cloud inference story [xwnm] Addressed? Partially (mostly reframed). Authors say they do not target real-time routing; instead they target periodic co-evolution where the SLM runs independently after distillation. This answers “routing missing” by changing scope, but leaves the realism/personalization critique largely outstanding.

11. Gains are small (~1–2%) and may be noise; effort vs payoff questionable [xwnm] Addressed? Weakly. Authors argue repeated runs and consistent deltas, and that small gains are meaningful at high baseline performance; skeptical reviewers may still consider this insufficient without stronger end-to-end or cost/benefit evidence.

12. Writing/story coherence and overclaims (notably flagged in discussion) [zRaQ, xwnm] Addressed? Partially. Authors acknowledge and upload multiple revised versions (blue/red highlights), remove privacy claims, clarify k=10, revise Section 4.4, and add setup details. Reviewer zRaQ still expresses frustration that revisions were initially superficial; authors later promise more substantial rewriting. Net: improved, but risk remains if final manuscript is still inconsistent.

**Reviewer Scores:**

Rsbi: likely 8 → 4 or 6: They were mostly impressed by their understanding that the framework provides a practical way to learn from new domain-specific data while preserving user privacy. However, the criticism of limited user privacy led the authors to remove privacy claims/

7Ggt: likely 4 → 4 Novelty concerns likely keep the 4.

zRaQ: likely 2 → 4 at best (only if the final revision truly fixes the story coherence and removes overclaims). Based on their tone, a change is possible but not guaranteed.

ZPEU: likely 6 → 4 given the limited contribution on communication overhead and enhanced privacy, which they highlighted as positive originally.

xwnm: likely 2 → 2 They explicitly retain their score post-rebuttal and seem unconvinced by both scope and experimental realism.

---

### Decision · Program_Chairs · 2026-01-26

Reject